# Finite element simulations for investigating the strength characteristics of a 5-m composite wind turbine blade

Can Muyan[1,2], Demirkan Coker[1,2*]

[1]Department of Aerospace Engineering, Middle East Technical University, Ankara, 06800, Turkey
[2]Structures and Materials Laboratories, RUZGEM (METUWIND) Center for Wind Energy Research, Middle East Technical University, Ankara, 06800, Turkey

*Correspondence to*: Demirkan Coker (coker@metu.edu.tr)

**Abstract.** Full-scale structural tests enable us to monitor the mechanical response of the blades under various loading scenarios. Yet, these tests must be accompanied by numerical simulations so that the physical basis of the progressive damage development can be better interpreted and understood. In this work, finite element analysis is utilized to investigate the strength characteristics of an existing 5-meter RUZGEM composite wind turbine blade under extreme flap-wise, edgewise and combined flap-wise plus edgewise loading conditions. For this purpose, in addition to a linear buckling analysis, Puck's (2D) physically based phenomenological model progressive damage analysis of the blade is performed. The 5-m RUZGEM blade is found to exhibit sufficient resistance against buckling. However, Puck's damage model indicates that laminate failure plays a major role for the ultimate blade failure. Under extreme flap-wise and combined load cases, the internal flange at the leading edge and the trailing edge are identified as the mainly damaged regions. Under edge-wise loading, leading edge close to root is the failure region. When extreme load case is applied as a combination of edgewise and flap-wise loading cases, less damage is observed compared to the pure flap-wise loading case.

## 1 Introduction

As fundamental eco-friendly renewable energy resources, wind turbines are designed to operate over a lifespan of 20 years. According to Holmes et al. (2007), long-term structural reliability of wind turbine components is vital when the high cost of manufacturing, inspection, and repair, especially for turbines located in remote regions, are considered. Composite blades are among the most critical components of a wind turbine, which are subjected to complex loading conditions. A rotor blade failure can have a significant impact on turbine downtime and safety. In order to assure sufficient mechanical resistance, structural testing and analysis must be conducted. However, structural testing methods, such as full-scale testing of the blade, are expensive and troublesome due to the construction of a test set-up. For a better understanding and interpretation of the the progressive damage development, tests are needed to be accompanied by numerical analysis methods (Chen et al., 2017). Moreover, structural analyses are utilized to calibrate structural blade test set-ups for different loading conditions.

In the literature, many studies on the structural behavior of composite turbine blades have been published in the past two decades with most of the studies conducted for large wind turbine blades. Jensen et al. (2006) carried out full-scale tests and nonlinear FE simulations of a 34-meter composite wind turbine blade under flap-wise loading. In both the tests and the simulations, they noted that spar cap (suction side) deflected nonlinearly and observed cross-sectional ovalisation, i.e. Brazier effect, arguing that Brazier effect is the main reason for the blade collapse. Overgaard et al. (2010) reported on testing and

numerical analysis of the collapse of a 25-meter composite wind turbine blade subjected to static flap-wise loading. In the study, in contrast to Jensen et al. (2006), the main failure mechanism which leads to blade collapse is argued to be delamination and its interaction with local buckling. It is stated that nonlinear strain behavior, i.e. local buckling is triggered by geometrical imperfections. According to Overgaard et al. (2010) delamination and local buckling occur prior to the Brazier effect observed in spar caps. Once the blade is weakened by the aforementioned failure mechanisms, blade collapses due to the compressive

strains in fiber directions.

     Yang et al. (2013) investigated the structural collapse of a 40 m blade and based on full-scale test results showed that debonding was the root cause of rotor blade collapse. Kim et al. (2014) studied the structural response of a Multi-MW class wind turbine blade using Puck's 3D damage model and linear buckling analysis. According to the results, blade shows sufficient structural strength and resistance against buckling. When laminate failure is of concern, the major weak point of the blade is located in

the skin at the maximum chord. In another study, Chen et al. (2014) conducted a full-scale bending test of a 52.3 m wind turbine blade. Delamination in the spar cap and shear web failure at the root transition region were found to be the main failure mechanisms for the blade collapse. Local buckling contributed to the main failure mechanism by facilitating local out-of-plane deformation. They conclude that for large blades, through-the-thickness stresses which cause debonding and delamination at the blade root transition region need to be considered in the FEA. In the follow-up study, Chen et al. (2017) stated that 3D

stresses/strains are important in the failure of a 52.3-meter blade and recommended use of solid elements in the FE simulation when debonding failure is of concern. 3-D strains and Yeh-Stratton failure criterion were used to calculate delamination and debonding failures in the blade utilizing sub-modeling technique.

     Haselbach and Branner (2016) discuss the initiation and development of trailing edge failure in the full-scale test of a 34-m wind turbine blade. They highlight the influence of buckling on the damage onset in the trailing edge and sandwich panel

failure. As a further outcome of the study, they show that modeling technique utilizing fracture mechanics approach for the failure in trailing edge delivers good agreement with experiments. Later, Haselbach (2017) investigated different trailing edge modeling methods in his work. He analyzed the trailing edge failure under edgewise and flap-wise loading conditions. He concluded that modeling the adhesive bonding in the trailing edge with solid brick elements and connecting them to the shell elements of the skin with Multi-Point Constraints (MPC) shows the best agreement with experiments. Recently, Castelos and

Balzani (2016) studied the effect of geometric nonlinearities on the fatigue analysis of the trailing edge bonding in wind turbine blades. They point out that the superposition of stresses for the fatigue may be misleading for modern, flexible rotor blades

where geometric nonlinearities must be considered. In addition to this, they propose a novel methodology for calculating stresses with a new load application method that reduces geometric nonlinear behavior of the blade.

Montesano et al. (2016), state that progressive failure models incorporating failure criteria do not consider the progressive nature of sub-critical microscopic intra-laminar damage, which is vital for predicting the onset of macroscopic failure modes. Therefore, in the study of a physically based multi-scale damage model, which can account for failures of a 33.25 m rotor blade under quasi-static and fatigue loading is introduced. The simulation results show the capability of the model to predict the evolution of sub-critical ply cracks between spar webs located near blade root at maximum chord length. The capability of the model to show damage evolution in the early stages of the progressive damage analysis is important for increasing damage tolerance accuracy and structural health monitoring. In the follow up work by Zu et al. (2018), the multi-scale model is further expanded to include cohesive zone elements to predict structural debonding failure at the spar/skin interface located near blade root at maximum chord.

In contrast to large wind turbine blades, a fewer number of investigations on the structural analysis of small wind turbine blades exists. Chen et al. (2015) focused on the local buckling resistance of 10.3 m wind turbine blades. FE analysis showed that configurations with sharp edges are susceptible to local buckling. During testing of the 10.3 m blade, although local buckling of shear web and flatback airfoil was observed, composite laminate failure in these locations was not observed. These results show the possibility of different failure mechanisms for different blade sizes. In another study, to improve the structural strength of a small size 9-m wind turbine blade, Paquette and Veers (2007) carried out a blade system design study (BSDS), where structural innovations such as flatback airfoils, thick root diameter and carbon spar caps were introduced and their advantages were demonstrated. The static strength of the blades was determined by measuring strains to failure by tests and using finite element analysis. Moreover, linear buckling analysis of the blades were conducted.

Fagan et al. (2016) presented a new iterative design process and utilized failure criteria to check the structural strength of different blade designs under various load cases. Besides this, Fagan et al. (2017) utilized failure criteria for the structural design optimization of a 13m glass-fibre epoxy composite wind turbine blade. In the study, experimental testing was used to calibrate finite element models. In these studies, Puck's damage model is used to determine the most suitable composite turbine blade design in terms of its structural behavior.

Within the framework of this study, the authors of this study refer back to their previous paper concerning the strength analysis of an existing 5-meter RUZGEM GFRP turbine blade using Puck failure criteria (Ozyildiz et al., 2018). As a part of the previous study, the linear Puck material model was compared with the progressive damage model (Puck). They concluded that progressive failure analysis is necessary to capture a more realistic simulation of failure mechanisms prior to testing.

The scope of this work is limited to the investigation of the structural response of an existing 5-meter wind turbine blade using global finite element modeling approach and progressive composite failure analysis. Hence, using the current modeling technique with shell elements, critical locations for failure and worst load case scenario are identified. Puck's 2-D damage

model demonstrates the direction to proceed for a complete and comprehensive modeling of the failure mechanisms. Furthermore, differences between edgewise, flap-wise and combined flap-wise/edgewise loading conditions are discussed. In combined edge-wise and flap-wise loading, less damage is observed compared to the pure flap-wise loading case.

The existing blade investigated in this work was designed as part of a joint-project between Core Team of the University of Patras and METUWIND (RUZGEM) – METU Center for Wind Energy. The blade was designed for a wind turbine that has 30 kW nominal power capacity at 10 m/s wind speed. According to the wind turbine characteristics, optimized aerodynamic blade design was finalized by the blade manufacturer. The existing blade consists of five main parts: suction side, pressure side, internal flange, "hat-shaped" chassis(spar), and flange, as seen in Figure 1.

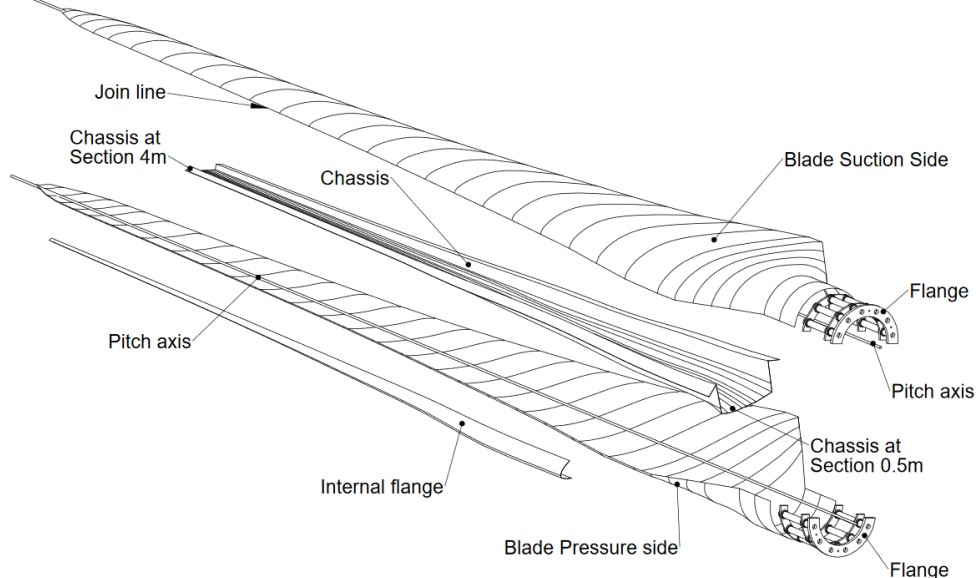

**Figure 1.** Blade assembly for the 5-meter RUZGEM turbine blade (Philippidis and Roukis, 2013).

## 2    Methodology

For the progressive failure analysis of the blade, Puck criteria explained briefly in the following paragraphs, are used. Puck failure criteria (Puck and Schuermann, 1998) are one the most commonly used and well-established criteria for the assessment of composite laminate strength. In this study, Puck's failure criteria are implemented for the evaluation of stress results of unidirectional and tri-axial composite laminates.

For fiber failure, Puck's criteria are as follows:

$$f_{E(FF)}^{T} = \frac{\sigma_1}{X_T} = 1 \text{ if } \sigma_1 > 0 \tag{1}$$

$$f_{E(FF)}^{C} = \frac{\sigma_1}{X_C} = 1 \text{ if } \sigma_1 < 0 \tag{2}$$

where $f^T_{E(FF)}$ and $f^c_{E(FF)}$ are stress exposures for fiber failure under tension and compression loading cases. $\sigma_1$ is the stress value in fiber direction, $X_T$ and $X_c$ are tensile and compressive strengths in fiber direction, respectively. Puck's inter-fiber failure uses different equations depending on the failure mode, which is detected. Under two dimensional (2-D) biaxial loading, the failure modes, which can be detected, are summarized in Figure 2 below. In Figure 2 the transition point from failure mode B to failure mode C is denoted by the point $(R^A_{\perp\perp}, \tau_{21c})$ and is calculated by the ratio $R^A_{\perp\perp}/\tau_{21c}$. Their values are calculated by the expressions below:

$$R^A_{\perp\perp} = \frac{S}{2p^{(-)}_{\perp\parallel}}\left[\sqrt{1 + 2p^{(-)}_{\perp\parallel}\frac{Y_c}{S}} - 1\right] \tag{3}$$

$$\tau_{21_c} = R_{\perp\parallel}\sqrt{1 + 2p^{(-)}_{\perp\perp}} \tag{4}$$

$$\text{and } p^{(-)}_{\perp\perp} = p^{(-)}_{\perp\parallel}\frac{R^A_{\perp\perp}}{S} \tag{5}$$

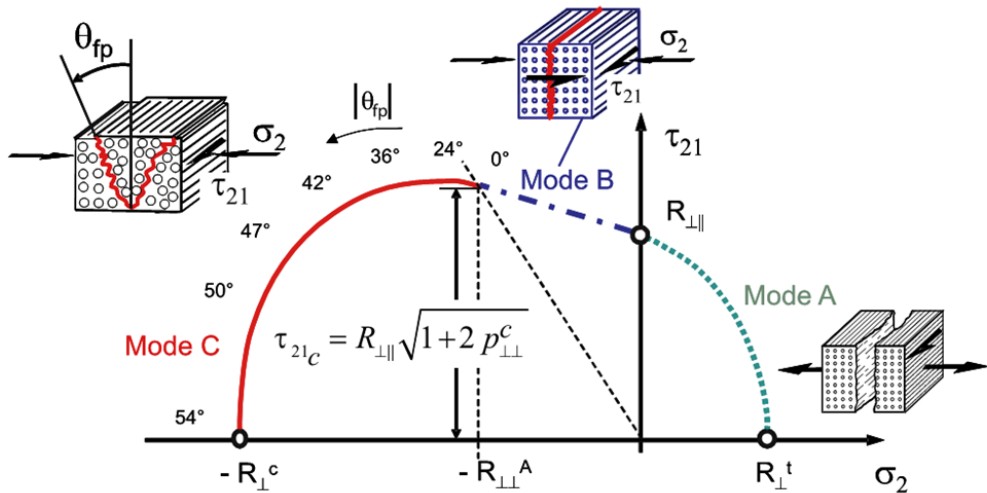

**Figure 2**. Failure envelope under biaxial loading (Knops, 2008).

Depending on the region of the failure envelope, the following inter-fiber failure expressions are written:

$$f^A_{E(IFF)} = \left[\left(\frac{\sigma_6}{S}\right)^2 + \left(1 - p^{(+)}_{\perp II}\frac{Y_T}{S}\right)^2\left(\frac{\sigma_2}{Y_T}\right)^2\right]^{1/2} + p^{(+)}_{\perp II}\frac{\sigma_2}{S} = 1 \text{ if } \sigma_2 \geq 0 \tag{6}$$

$$f^B_{E(IFF)} = \frac{1}{S}\left\{\left[(\sigma_6)^2 + \left(p^{(-)}_{\perp II}\sigma_2\right)^2\right]^{1/2} + p^{(-)}_{\perp II}\sigma_2\right\} = 1 \text{ if } \begin{cases} \sigma_2 < 0 \\ \left|\frac{\sigma_2}{\sigma_6}\right| \leq \frac{R^A_{\perp\perp}}{\tau_{21C}} \end{cases} \tag{7}$$

$$f_{E(IFF)}^C = \left[\left(\frac{\sigma_6}{2(1+p_{\perp\perp}^{(-)})S}\right)^2 + \left(\frac{\sigma_2}{Y_C}\right)^2\right]\frac{Y_C}{(-\sigma_2)} = 1 \text{ if } \begin{cases} \sigma_2 < 0 \\ \left|\frac{\sigma_2}{\sigma_6}\right| \geq \frac{R_{\perp\perp}^A}{\tau_{21C}} \end{cases} \tag{8}$$

In the equations above $p_{\perp II}^{(+)}, p_{\perp II}^{(-)}$ and $p_{\perp\perp}^{(-)}$ represent inclination parameters that control the shape of the failure envelope. According to Puck and Schuermann (1998), $p_{\perp II}^{(+)} = 0.3$ and $p_{\perp II}^{(-)} = 0.25$ are chosen for the GFRP material. $\sigma_2$ is the stress value in the transverse fiber direction, $Y_T$ and $Y_c$ are tensile and compressive strengths in the transverse fiber direction. Shear
stress and shear strength are represented by $\sigma_6$ and S, respectively. If the value of stress exposure ($f_E$) exceeds 1, failure initiation occurs. Mode A is caused by tensile and shear stresses. Modes B occurs under compressive and shear stresses. Mode C is a dangerous failure mode in compressive shearing, which may lead to delamination.

Degradation rules are applied to the elements which fail according to the specific Puck's failure criteria that are inter-fiber failure (IFF) mode A, B, or C (Eq. (6), (7) and Eq. (8), respectively). As presented by Passipoularidis et al. (2011), based on
degradation rules in Table 1, transverse elasticity and shear moduli of the damaged elements are reduced accordingly. Recommended parameters c, $\eta_r$ and $\xi$ for the degradation function of the GFRP material in Eq. (8) are taken from Knops and Bögle (2006). $f_{E(IFF)}^A$, $f_{E(IFF)}^B$ and $f_{E(IFF)}^C$ are the stress exposure values that are considered for determining the failure mode during the analysis.

**Table 1:** Degradation rules according to the failure mode.

| Failure Mode | Degradation Rule |
|---|---|
| FF (Tension / Compression) or IFF(C) in at least 1/3 plies of a laminate | Failure of the laminate |
| IFF (A) | $E_2 = \eta E_2$ <br> $G_{12} = \eta G_{12}$ <br> $\nu_{12} = \eta \nu_{12}$ |
| IFF (B) | $E_2 = \eta E_2$ <br> $G_{12} = \eta G_{12}$ <br> $\nu_{12} = \eta \nu_{12}$ |
| IFF (C) | $E_2 = 0.1 E_2$ <br> $G_{12} = 0.1 G_{12}$ <br> $\nu_{12} = 0.1 \nu_{12}$ |

In Eq. (8) $\eta$ is known as the degradation factor and can be expressed according to the equation below:

$$\eta = \frac{1-\eta_r}{1 + c(f_{E(IFF)} - 1)^\xi} + \eta_r \tag{9}$$

The summary of the algorithm of the FE analysis based progressive failure analysis of a composite laminate using Puck failure criteria is shown in **Figure 3**. The complete algorithm is implemented using ANSYS Parametric Design Language (APDL). First, using ANSYS APDL script, different material numbers are given to each lamina, which constitutes a layer of a composite laminate. This step is necessary, because during the execution of the progressive damage propagation, each lamina is subjected to different degradation rules. Then, an extreme load case is applied incrementally to the model, and static analysis is run. Afterward, in the post-processing module, stresses are read. From Puck failure criteria for FF (Eq. (1) and (2)) and IFF (Eq. (6), (7) and (8)), stress exposures are calculated. Depending on the rules presented in Table 1whether ply failure happens is checked. If ply failure occurs due to First-Fiber-Failure (FFF) or if IFF(C) is observed in at least one third of the plies of a laminate, laminate failure is assumed to take place. If IFF A or B or IFF C in any ply is seen, gradual degradation rules are applied, and after the assembly of the new constitutive material model, the load is increased, and the analysis is re-run. This calculation procedure is run until the solution does not converge. If the solution does not converge, the analysis aborted. As seen from the flow chart, as long as no FF or IFF failure occurs, without updating the constitutive material model, load is incremented, and the analysis is executed.

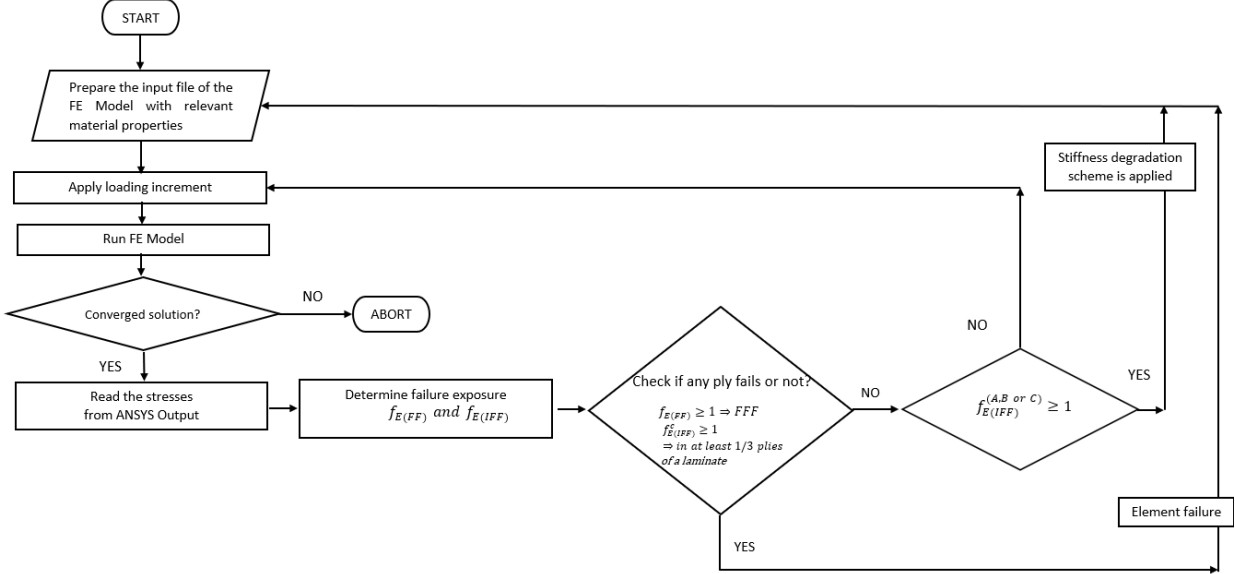

**Figure 3.** Flowchart of the FE Analysis based strength analysis of a composite laminate using Puck's damage model.

Validation of the APDL Code against experimental data provided by (Hinton et al, 2002) is done for the $[0/90]_s$ GFRP laminate/MY750 and $[0/\pm45/90]_s$ CFRP laminate/AS4 3501-6 under uniaxial tension loading. As seen from **Figure 4**, there is a good agreement between APDL Code and experimental data for the progressive failure analysis of $[0/90]_s$ GFRP/MY750 and $[0/\pm45/90]_s$ CFRP/AS4 3501-6 laminates.

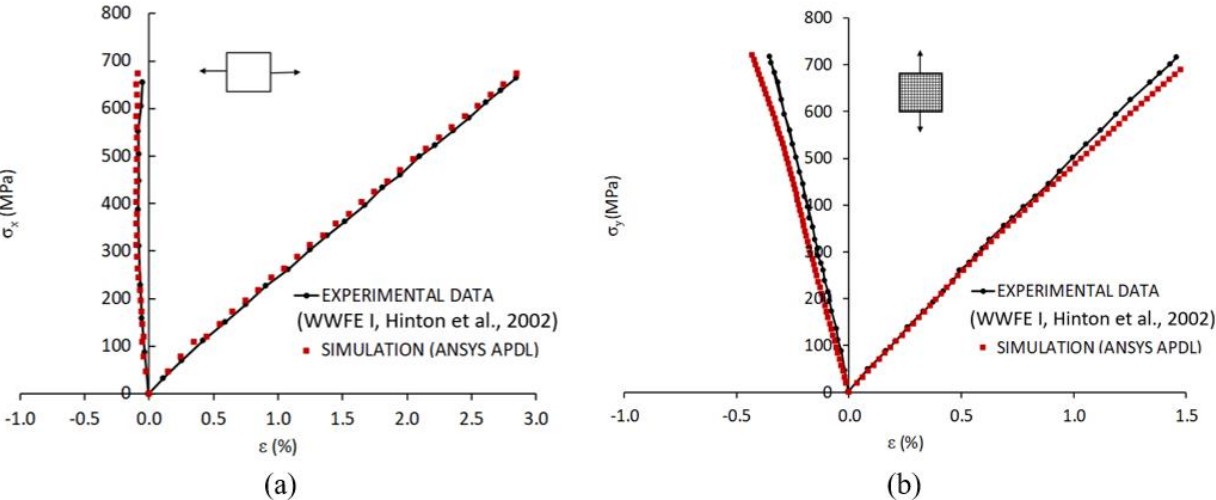

**Figure 4.** Validation of the APDL Code for the progressive failure analysis of (a) $[0/90]_s$ GFRP/MY750 laminate under $\sigma_x$ uniaxial tension (b) $[0/\pm45/90]_s$ CFRP/AS4 3501-6 laminate under $\sigma_y$ uniaxial tension.

## 2.1 Finite Element Model of the RUZGEM Blade

The 2D blade technical drawings, which include the blade aerodynamic design details such as chord length and twist angle along the blade, were provided by the blade manufacturer Compblades. By using these given 2D blade drawings, the 3D CAD model of the blade is prepared in NX 12.0 environment. In Table 2, the material properties and design allowables of the blade materials for static analysis is listed (Philippidis and Roukis, 2013). Referring to Germanischer Lyod (2010), design allowables are obtained from the knockdown of the experimental strength values by the material safety factor 2.406.

**Table 2.** Material properties and design allowables of RUZGEM blade for static analysis

| Material Property | | Unidirectional Laminate | Steel | Gel Coat | CSM 300 | Divinycell H45 |
|---|---|---|---|---|---|---|
| **Density, ρ** | [kg/mm³] | 1896 | 7850 | 1200 | 1896 | 200 |
| **Thickness, h** | [mm] | 0.716 | 5.3 | 0.9 | 0.358 | 5 or 10 |
| **$E_1$** | [GPa] | 24.84 | 210 | 3.98 | 9.14 | 55x10⁻³ |
| **$E_2$** | [GPa] | 9.14 | | | | 55x10⁻³ |
| **$v_{12}$** | | 0.29 | 0.3 | 0.34 | 0.29 | 0.4 |
| **$G_{12}$** | [GPa] | 2.38 | | | | 15x10⁻³ |
| **$X_T$** | [MPa] | 191.73 | 581.8 | 35.29 | 16.86 | 1.4 |
| **$X_C$** | [MPa] | 101.16 | | | | 0.6 |
| **$Y_T$** | [MPa] | 16.86 | | | | 1.4 |
| **$Y_C$** | [MPa] | 50.41 | | | | 0.6 |
| **S** | [MPa] | 11.29 | | | | 0.56 |

The skin of the blade is composed of unidirectional and tri-axial laminates, whereas only tri-axial laminates are used for the spar. The lay-up sequence for the pressure and suction side differs only in the area from 1.25m to 2.0m, where an extra unidirectional glass fabric was placed in the suction side of the blade. The root part of the blade is composed of unidirectional laminate, tri-axial laminates, and steel. The outer surface of the blade is covered with transparent Gel Coat and a layer of chopped strand mat, 300 $g/m^2$ CSM 300. In addition to this, the Divinycell H45 foam used in the trailing edge is of 10 mm thickness in the area from 0.7m to 2.0m and 5mm thickness from 2.0m to 3.0m. Since the gel coat, CSM 300, and foam do not have a significant contribution to the strength of the blade; these materials are not included in the finite element model. The details of the material lay-up and geometry are given in the supplement.

After geometric modeling of the blade, the material model of the blade structure is prepared in Ansys ACP/Pre module. Plane stress SHELL 181 quadrilateral elements are used to mesh the blade entirely in ANSYS Workbench, as seen in Figure 5. The mesh density of the FE Model and the boundary conditions are depicted in **Figure 5**. For the blade FE model all rotational and translational degrees of freedom at the blade root are fixed. A total number of approximately 35,000 nodes are used in the finite element model, which is determined based on the mesh convergence study discussed later in this subsection. SHELL 181 elements with size 15x15 mm are used for the mesh. This element size correlates to the typical element size for small scale wind turbine blades. Adhesive materials are used for connections: pressure side - spar - suction side, pressure side - internal flange - suction side along the leading edge and finally between suction side - pressure side. These connections are modeled using bonded contact with Augmented Lagrange algorithm in the FE Model. Different trailing edge modeling methods as presented in Haselbach (2016) can be implemented and compared with experimental data in the future.

Since the total displacement of the blade is relatively small compared to the total length of the blade under 100% flap-wise, edgewise and combined loading cases, the analysis is limited to linear geometry. Another reason for the choice of linear geometry option is to avoid convergence problems and excessive computation time at higher load levels. In addition to this, using the global modeling of the 5-m wind turbine blade with plane stress elements, through-the-thickness stresses, which are necessary for a detailed examination of delamination and/or debonding failures cannot be obtained.

Load displacement curves are utilized for investigating the structural response of the blade under various loading scenarios. In Figure 6 load and displacement measurement points are displayed. Loads are measured at the blade pitch coordinate system defined at the root center according GL Guidelines 2010 and displacements are measured from point P at the spar tip, which is highlighted in green. The exact coordinates of the displacement point with respect to the coordinate system at blade root are x=-9.801, y=-4.15, z=4000.

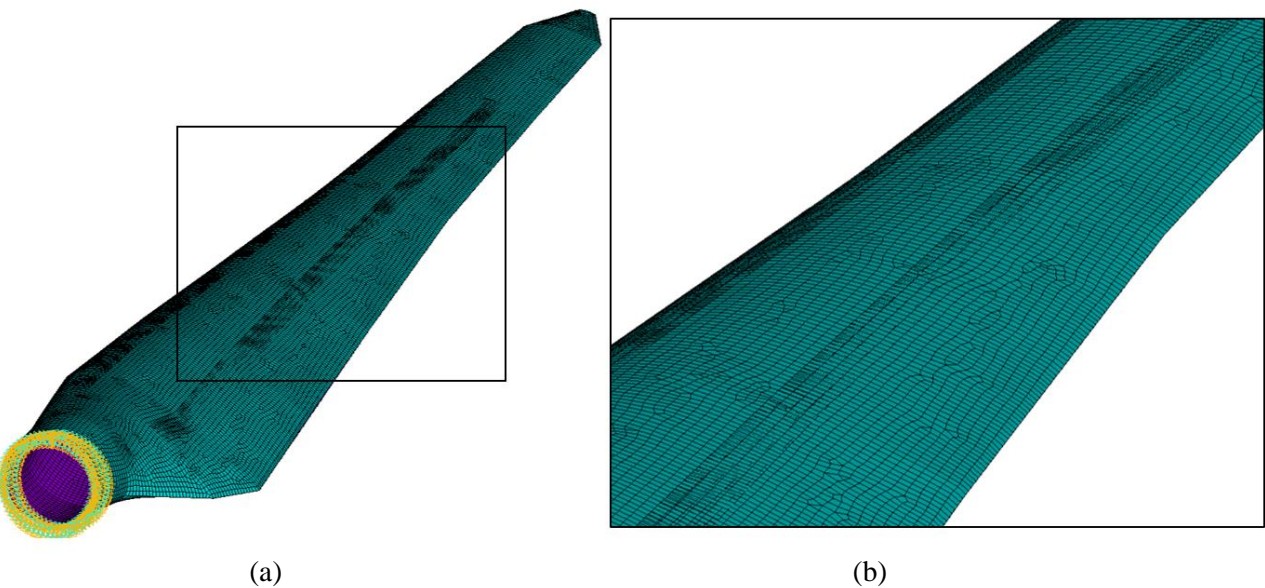

|   |   |
|---|---|
| (a) | (b) |

**Figure 5.** (a) Boundary conditions and the mesh density of the FE Model, (b) blade detail.

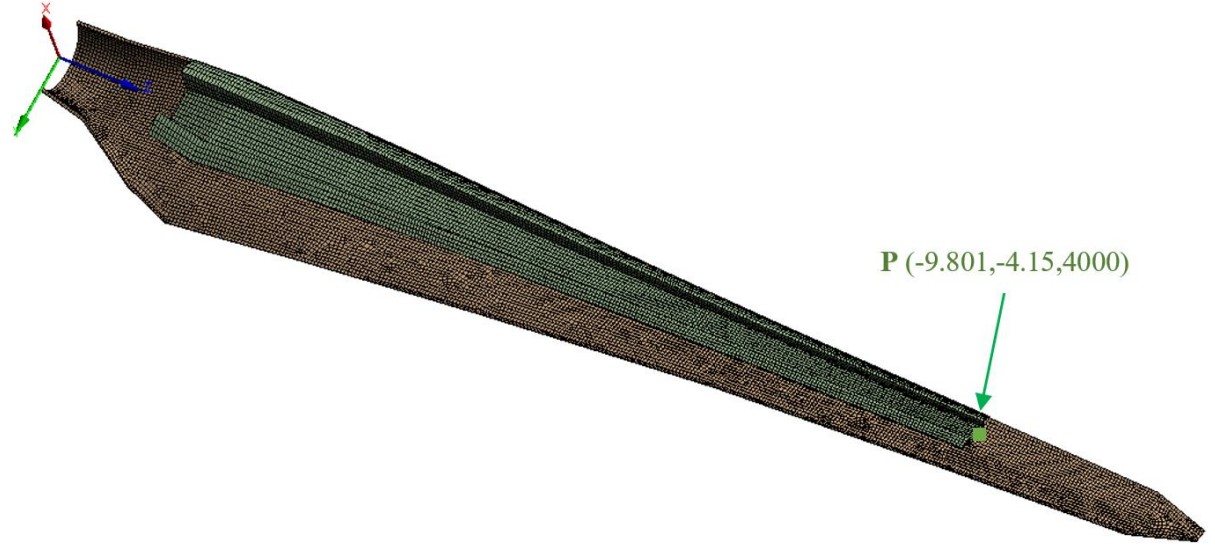

P (-9.801,-4.15,4000)

Figure 6. Location of the load measurement coordinate system and the deflection measurement point P.

For the design of the RUZGEM blade, the turbine specifications are obtained from the meteorological data in Ankara, Turkey (Weinzierl and Pechlivanoglu, 2013). These were analyzed so that average wind speed, the occurrence of gusts, and wind speeds are determined. Based on this information, the turbine specifications were selected according to British Standard IEC 61400-2 standard (2006). Loads for the structural design were selected as the worst load case scenario for the complete set of IEC extreme loads provided from aero-elastic simulations performed at Smart Blade GmbH (Weinzierl and Pechlivanoglu, 2013).

Mixture of different Design Load Cases (DLCs) and time instances are used for this study and moments are extreme for all positions along the blade. The extreme loads were computed using the wind turbine aero-hydro-servo-elastic software tool FAST version v7.01.00a-bjj. During the simulations, the turbine is simulated as a stall regulated constant speed turbine at 83 rpm with a gearbox and simple induction generator. Using this input, the blade is analyzed under extreme loads in the flap-wise and edge-wise directions. Loads are calculated at 28 stations along the blade span direction.

According to the recommendations of IEC 61400-23:2002(2015) standard a partial safety factor of 1.35 is included in the loads for the FE simulations. The forces and moments are given in the blade pitch coordinate system in **Figure 7** according to GL Guidelines 2010, where Fx, Fy, Fz are the forces in flap-wise, edgewise and axial directions. $M_X$, $M_Y$, $M_Z$ are the moments in edgewise, flap-wise and pitch directions. After considering partial safety factors, extreme flap-wise and edge-wise external load distributions are plotted along the spar of the blade length in **Figure 8**. Similarly, **Figure 9** shows the calculated values

of the flap-wise and edge-wise bending moment in sections along the blade span length. These edgewise and flap-wise moment values are computed from flap-wise and edge-wise loads, respectively. As stated in IEC 61400-23 standard, for the FE simulations, external flap-wise and edge-wise loads are increased incrementally by 1% of extreme load case until the collapse of the blade occurs.

    Extreme external loads are given over 28 stations of the blade suction and pressure sides as displayed in Figure 10(a) for flap-

225 wise loading and Figure 11(a) for edgewise loading. The loads at these stations are distributed among the nodes along the spar width on the suction and pressure sides as shown in Figure 10(b) and Figure 11(b) for flap-wise and edgewise load applications, respectively.

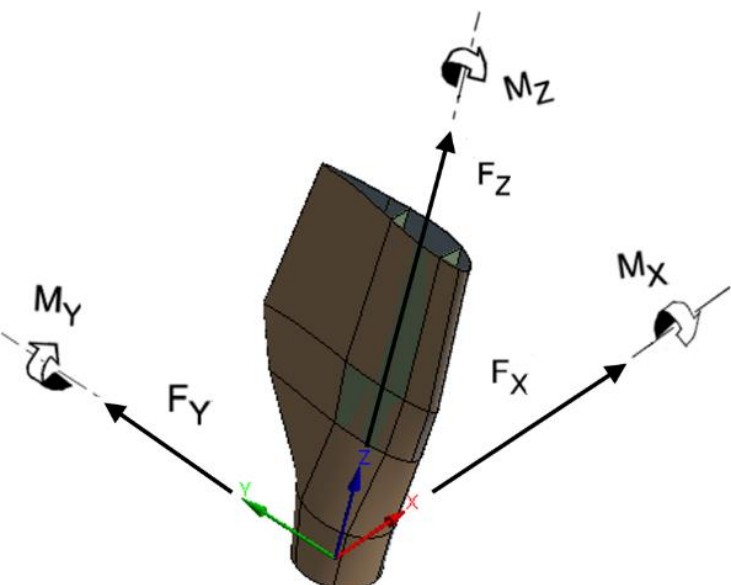

**Figure 7.** Blade pitch coordinate system (GL Guidelines 2010).

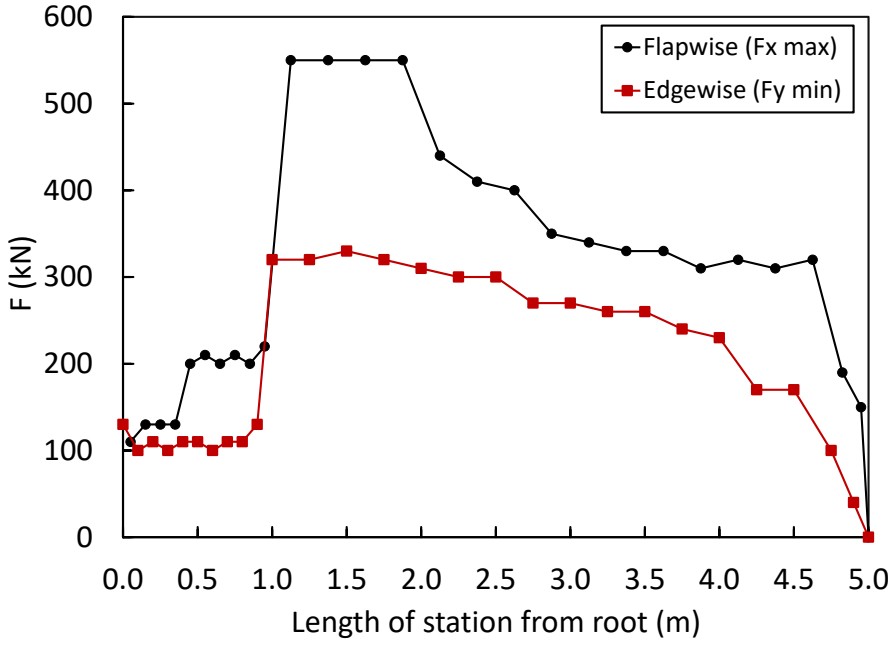

**Figure 8.** Extreme external load distribution along span length.

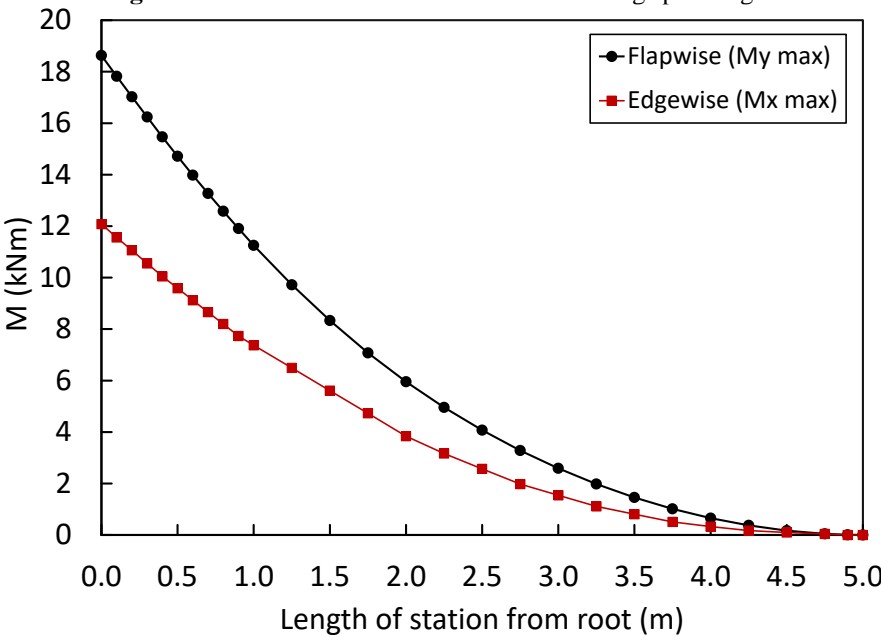

**Figure 9.** Extreme moment distribution along span length.

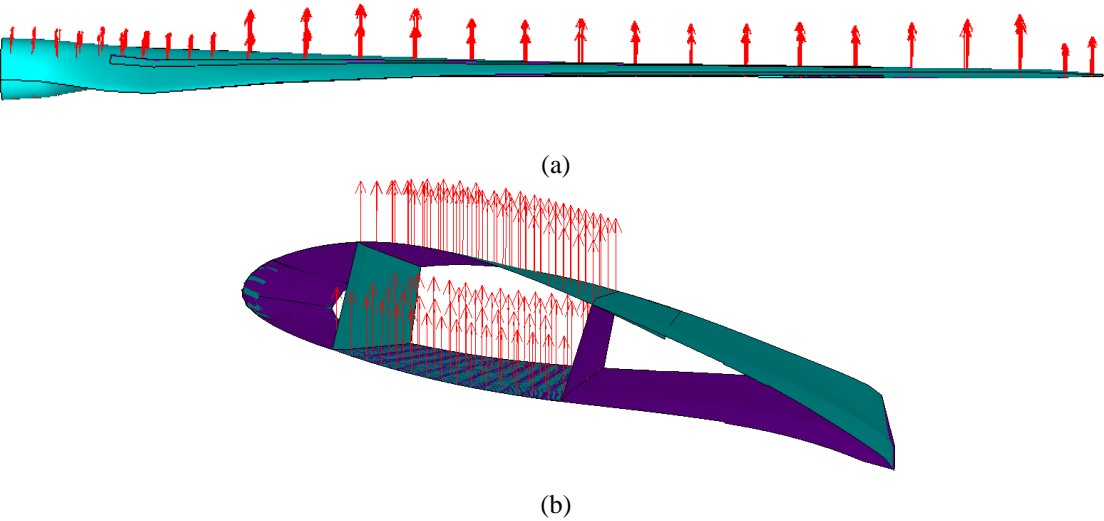

(a)

(b)

**Figure 10.** Extreme flap-wise load application to blade FE Model: (a) side and (b) cross-section view.

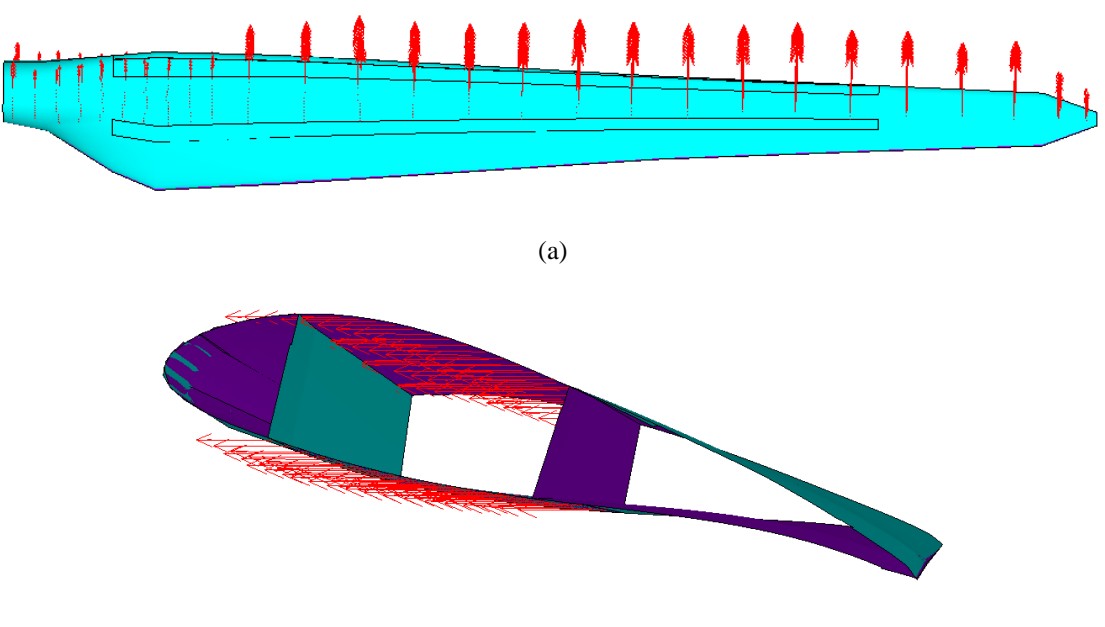

(a)

(b)

**Figure 11.** Extreme edge-wise load application to blade FE Model: (a) top and (b) cross-section view.

In order to decide for the reasonable mesh density, mesh convergence study is conducted. Mesh convergence is shown for total displacement at blade tip under extreme flap-wise loading and first eigenfrequency as seen in **Figure 12** (a) and (b) respectively. There are of approximately 35,000 nodes in the FE Model. The decision for the mesh refinement is based on a good compromise between accuracy and computing time. The analysis takes about thirty-six hours for flap-wise and combined load cases, whereas for edgewise loading around seventy-two hours are needed with an Intel Xeon E-5-1620 v3 Workstation PC with 64 GB RAM.

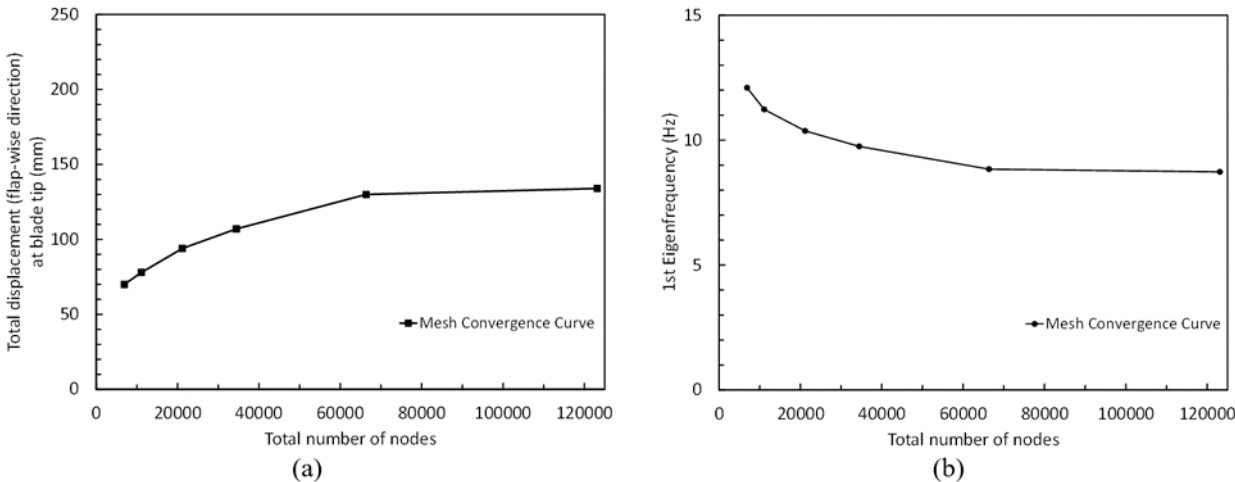

**Figure 12.** Mesh convergence study using (a) total number of nodes vs. total displacement (flap-wise direction) at blade tip under flap-wise loading (b) total number of nodes vs. 1. Eigenfrequency.

## 3 Results

This section begins with a linear buckling analysis, followed by the progressive failure analysis of the blade under extreme flap-wise, edgewise and combined loading conditions. This section is followed by a discussion regarding the comparative study of these three loading scenarios.

### 3.1 Buckling Analysis

Linear buckling analysis of the blade is performed in order to investigate its buckling resistance and location of buckling eigenmodes. The results are depicted in Figure 13, which shows the buckling modes of the blade under (a) 100% flap-wise(max) loading case (Figure 12a) 100% edgewise(min) loading case (Figure 12b) and 100% combined edgewise(min) and flap-wise(max) loading (Figure 12c). Negative buckling factors correspond to the loads applied in the opposite direction, because no critical eigenvalue could be found in the load application direction. In other words, the blade exhibits sufficient buckling resistance in the load application direction for flap-wise, edgewise and combined edgewise plus flap-wise loading. According to GL 2010 the load factor should be greater than 1.25, which is fulfilled for all the load cases studied. We note that in other cases in the literature such as Paquette and Veers (2007) and Chen et al. (2015), local buckling of small size wind turbine blades is found to be a major design concern.

Location of buckling regions are observed in the suction side and pressure side trailing edges for flap-wise, combined edgewise and flap-wise loading cases, respectively. For the extreme edgewise load case, buckling failure location is in the pressure side trailing edge towards the blade root. The buckling mode locations shows similarity with buckling mode locations of the multi-MW blade in the study of Kim et al. (2014) and blade configurations with PVC foam in the study of Chen et al. (2014) as they are located in the wide trailing edge region near the maximum chord. However, in the study of Paquette and Veers (2007), buckling failure location is in the root transition region.

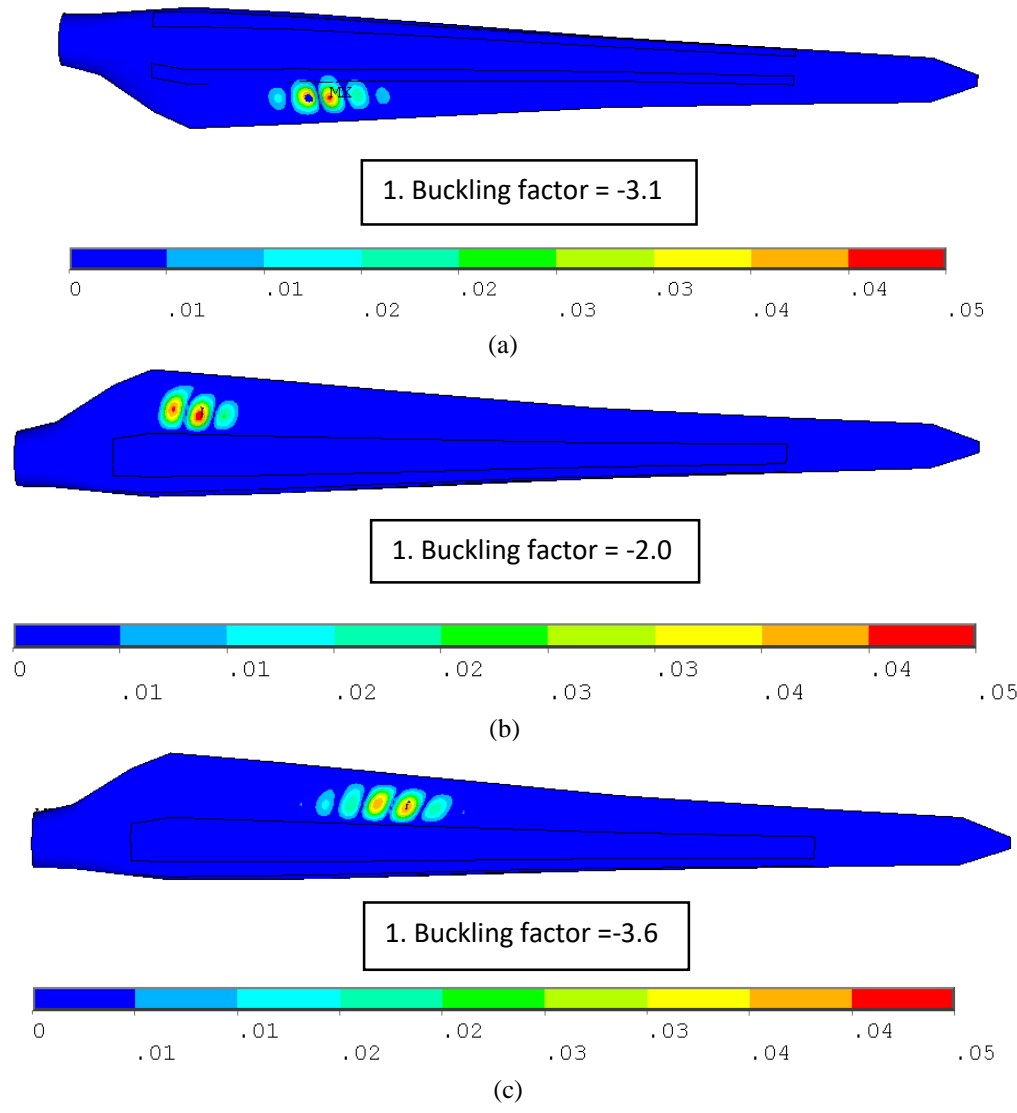

1. Buckling factor = -3.1

(a)

1. Buckling factor = -2.0

(b)

1. Buckling factor =-3.6

(c)

Figure 13. Buckling modes of the blade under (a) 100% flap-wise(max) loading case (b) 100% edgewise(min) loading case (c) 100% combined edgewise(min) and flap-wise(max) loading. (Color bar shows total deformation).

### 3.2    Progressive Damage Analysis

In this subsection results, from the progressive damage analysis of the RUZGEM blade subjected to flap-wise, edgewise and
combined loading conditions are presented.

#### 3.2.1    Progressive damage analysis under flap-wise (max) loading

The total deformation of the nonlinear blade model versus the undeformed model under 100% extreme flap-wise loading is
displayed in Figure 14. The maximum blade deflection at the blade tip is 121 mm.

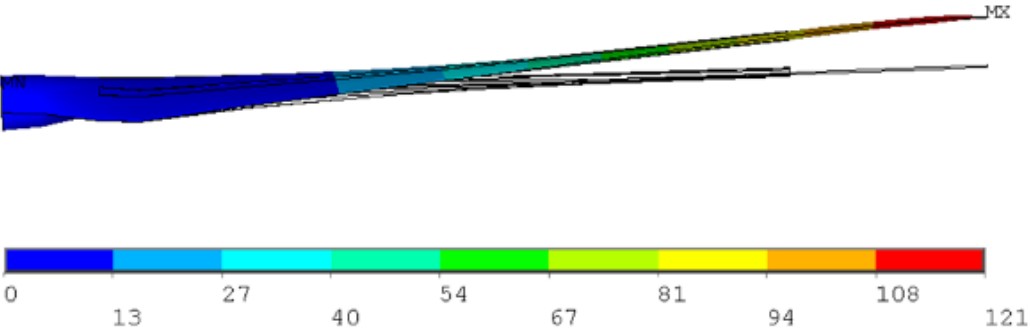

0    13    27    40    54    67    81    94    108    121

Load displacement curves of the blade in the range between 10% - 120% of extreme flap-wise loading are displayed for the linear elastic model and progressive damage model (Puck) in Figure 15. Load application and displacement measurement is done in the flap-wise direction. Based on this load displacement curve, element failure progression in the damaged blade components; pressure side, internal flange and suction side at 75%, 100%, 105% and 116% is depicted in **Figure 17** below. According to the implemented methodology, an element fails if FF or IFF(C) in at least one third of the plies of a laminate is detected. It can be observed from the figure that up to 75% of the extreme flap-wise loading, the stiffness for both linear elastic and progressive damage models remains almost the same. However, the analysis output data show that degradations in the transverse elasticity, shear moduli and Poisson's ratio starting from 13% loading occur, but this does not play a significant role in the deflection of the blade as can be seen from Figure 15. According to the analysis results, element failure is observed in the internal flange at 75% loading. This is the first slope drop of the load displacement curve and can be considered as the first turning point. As seen from **Figure 17**, failure in the internal flange grows further as the load is increased to 100%. At 105% loading in addition to the damaged region in the internal flange, damage grows along the trailing edge and leading edge. This turning point can be regarded as the second slope drop in the load displacement curve. Due to the element failure, deformation in the form of local buckling at the trailing edge is observed in Figure 16 below.

Shortly before collapse at 116% loading, damaged regions at the leading and trailing edges evolve further and damage close to the blade tip occurs. The reasons for damage initiation at the blade tip at the most extreme load level can be explained by the fact that there is although low, some loading on the blade tip as seen in **Figure 8**. At 116%, 1.16 times the extreme flap-wise loading, which read is from **Figure 8** is applied to the blade and the blade collapses afterwards. Moreover, blade tip structure is rather thin and less stiff compared to other regions of the blade. On top of this, as seen in **Figure 17** (d) at 116% load level, trailing edge and the internal flange which is used to bond the pressure and suction sides of the blade are already damaged. As a consequence, towards the blade tip the pressure and suction sides of the blade are detached at this load level. Under these circumstances, blade tip structure is weaker and can be damaged more easily. Likewise, debonding of suction and

pressure sides from the adhesive joints was reported as the main failure mechanism causing a progressive collapse of the blade
structure in Yang et al.(2013). Finally, the blade collapses after 116% of extreme flap-wise loading.

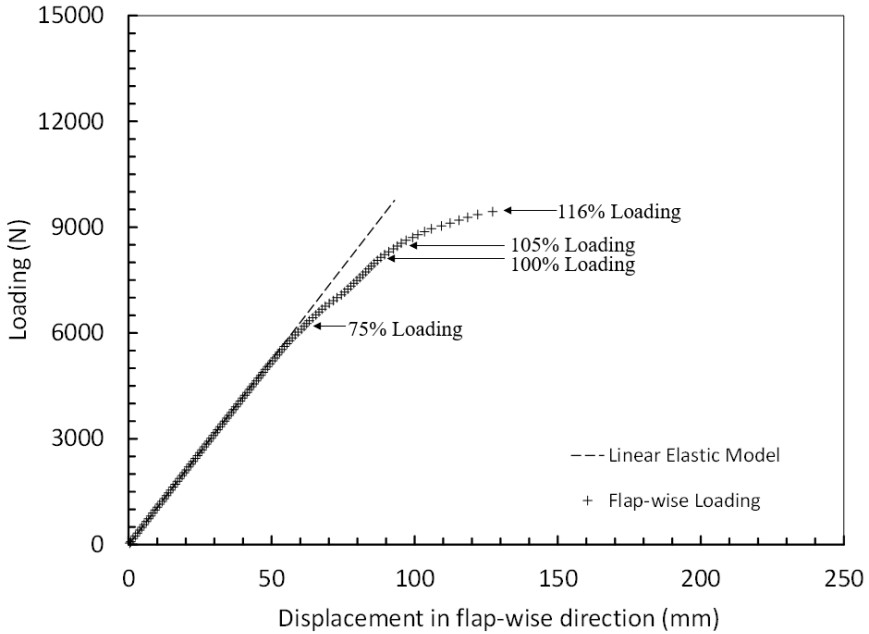

**Figure 15.** Load displacement curves of the blade using the linear elastic model and progressive damage model (Puck) under extreme flap-wise loading.

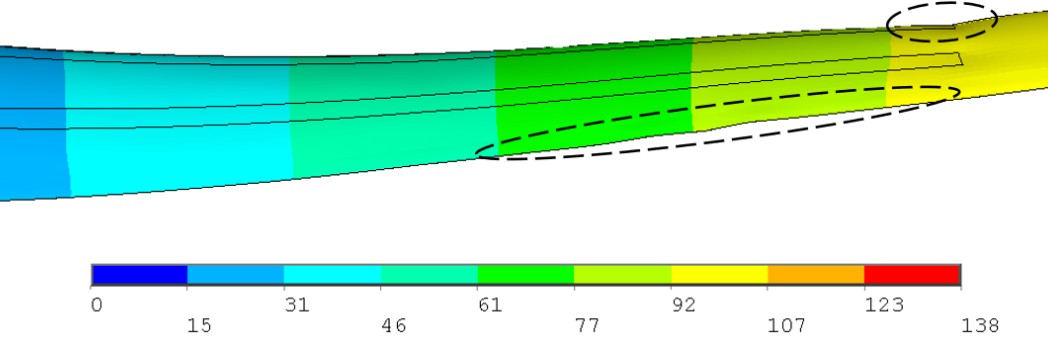

Figure 16. Total deformation occurring in the form of local deformation due to failed elements at 105% flap-wise(max) loading. (Scale factor for deformation plot: x20).

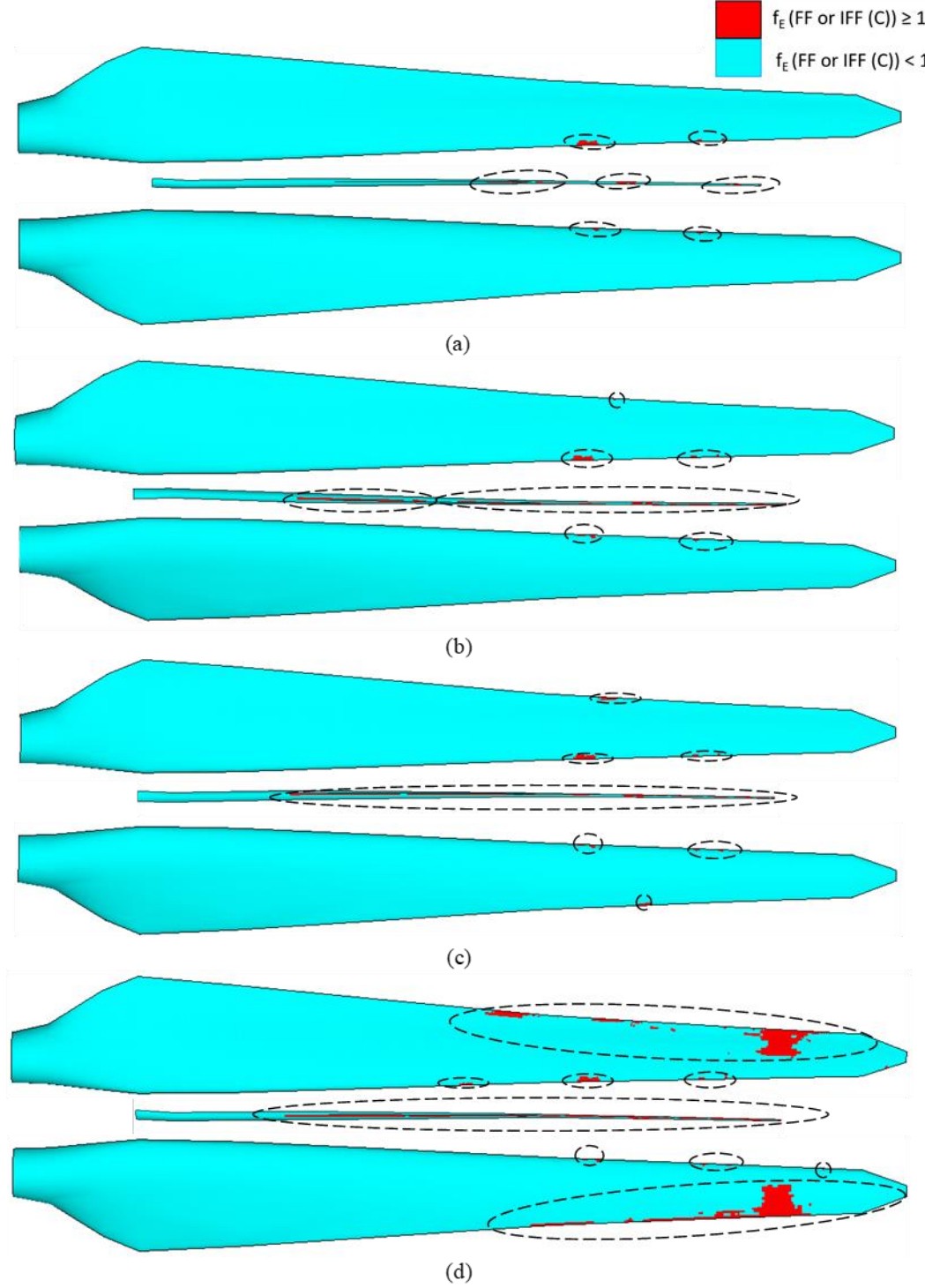

Figure 17. Element failure progression in the pressure side, internal flange and suction side of the blade (from top to bottom in a row) at (a) 75%, (b) 100%, (c) 105% and (d) %116 of extreme flap-wise loading.

In Figure 18 inter-fiber failure mode A (IFF(A)) and inter-fiber failure mode B (IFF(B)) distribution in pressure side, internal flange and suction side of the blade at 50% flap-wise loading are shown. If both stress exposures are present in an element, the higher stress exposure IFF(A) or IFF(B) is shown. Based on the output data from the FE analysis of the blade, the damage initiation begins at 13% of extreme flap-wise loading due to IFF(A) or IFF(B). As discussed in the methodology chapter, stress exposures greater than or equal to one indicate damage, and damaged regions are shown in red. The figure shows that inter-fiber failure is present in the internal flange and the trailing edge of the blade. It is worth noting that, IFF(A) and IFF(B) do not lead to the element failure. When IFF(A) or IFF(B) occurs, only the transverse, shear moduli, and Poisson's ratio are reduced according to the degradation rules. It is noted that FF and IFF(C) initiate in the same location as IFF(A) and IFF(B), which can be considered as subcritical cracks. Similar to our findings, Montesano et al. (2016) states that (referring to Lambert et al. (2012) and Sorensen et al. (2004) studies) subcritical cracks can act as precursor to more critical damage modes such as delamination or adhesive debonding. In this study, IFF(C), which is a dangerous failure mode indicating risk of delamination and FF are the critical failure modes which lead to element failure. Recently, Chen et al. (2019) reported that matrix-dominant failure and delamination occurs before fiber-dominant failure at the trailing edge. Likewise, we observe IFF(A) or IFF(B) failure before FF at the trailing edge.

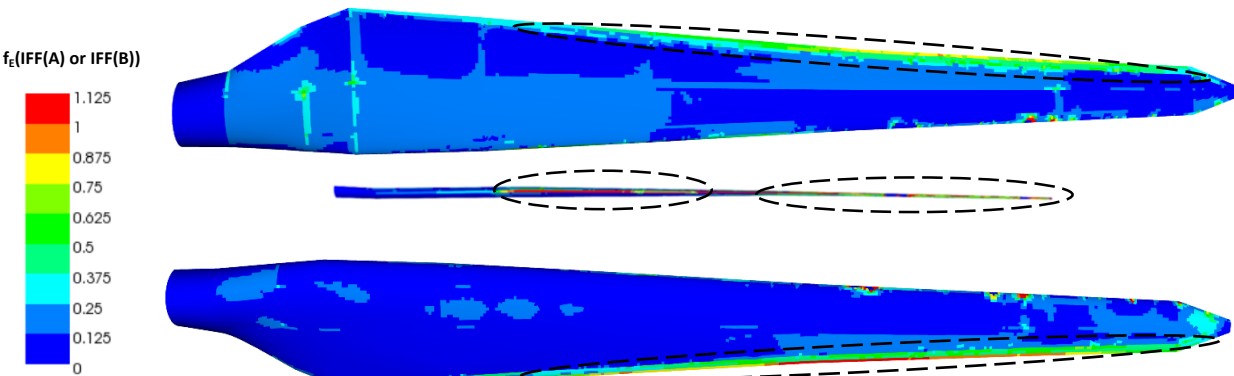

**Figure 18**. Damage evolution (IFF(A) or IFF(B)) in the pressure side, internal flange and suction side of the blade (from top to bottom in a row) at 50% of extreme flap-wise loading.

### 3.2.2    Progressive damage analysis under edgewise (min) loading

The total deformation of the nonlinear blade model versus the undeformed model under 100% extreme edgewise loading is displayed in Figure 19. The maximum blade deflection at the blade tip is 31 mm and much less than the deformation compared to pure flap-wise loading.

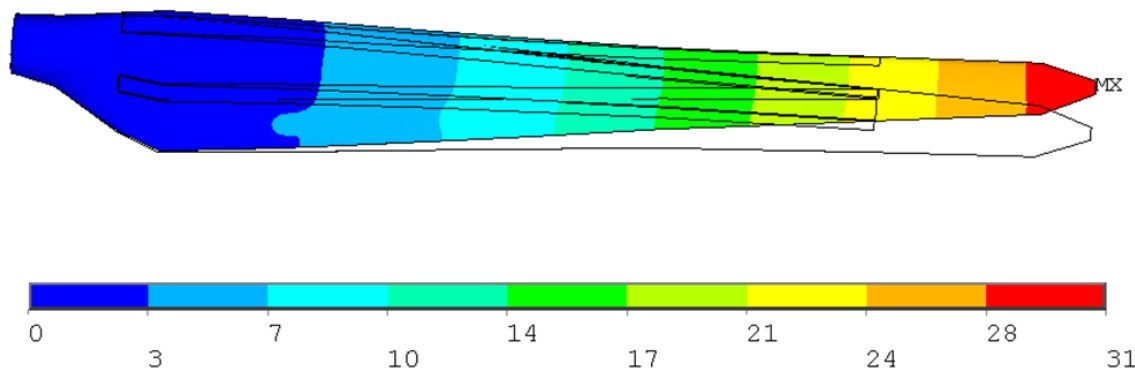

**Figure 19.** Total deformation of the RUZGEM blade model vs. undeformed model under 100% extreme edge-wise loading condition (scale factor: x8).

Load displacement curves in the range between 10% - 320% of extreme flap-wise loading of the blade are displayed for the
linear elastic model and progressive damage model (Puck) in Figure 20. In the figure, the load is applied and displacement at the spar tip is measured in the edgewise direction. Figure 22 shows the element failure evolution in the blade components; pressure side, internal flange and suction side of the blade at 100%, 210%, 309%, and 314% load levels. Stress exposures fiber-failure (FF) and inter-fiber failure mode C (IFF(C)) distribution are shown on the same plot. Since element failure in spar is not detected within above mentioned loading range it is not shown in Figure 22. The regions where the stress exposure is
equal to or greater than one is depicted in red. The stiffness of these failed elements is set to zero and they do not contribute to blade strength anymore. It is seen that at 100% loading element failure is not detected. It can further be observed from the figure that up to 210% of the extreme flap-wise loading, the stiffness for both models remains almost the same. After 210% loading, the stiffness is reduced slightly due to the element failures at the internal flange tip (See Figure 20). At %309 loading element failure at the leading in blade root is observed. Due to edgewise (min) loading, compressive stresses are dominant in
this region and the blade material is weaker in compressive than in tension. This failure causes the more significant slope drop in the curve and at 314% loading the blade is close to collapse. At the turning point which corresponds to 309% edgewise loading, the stiffness of the failed elements in the leading edge towards blade root are set to zero and total deformation due element failure in the form of local buckling is observed as depicted Figure 21 below.

Based on the results, blade design exhibits excessive safety in edgewise direction and is considered to be over-conservative
for this type of loading.

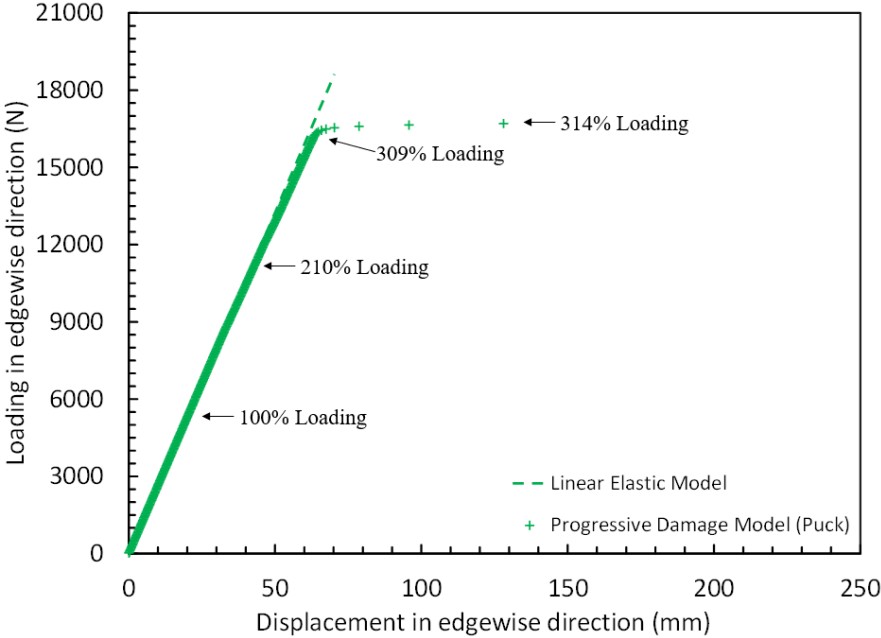

**Figure 20.** Load displacement curves of the blade using the linear elastic model and progressive damage model (Puck) under extreme edgewise loading

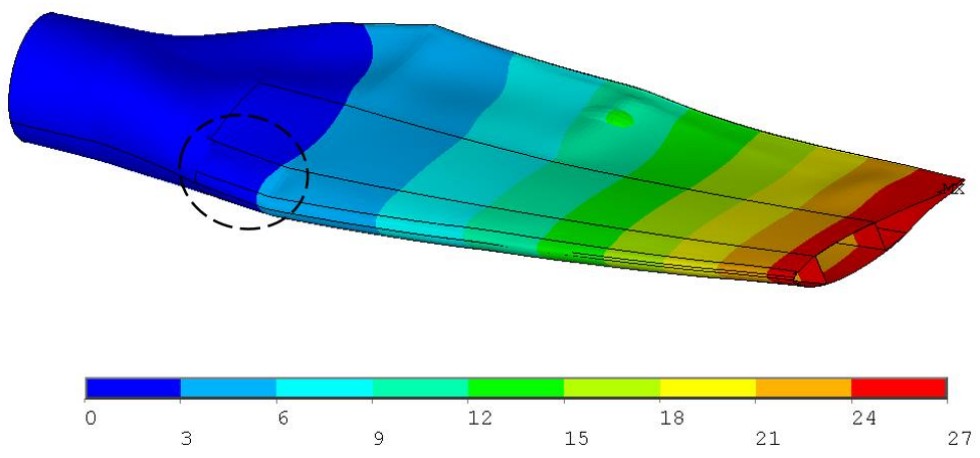

Figure 21. Total deformation occurring in the form of local deformation due to failed elements at 309% edgewise(min) loading. (Scale factor for deformation plot: x15).

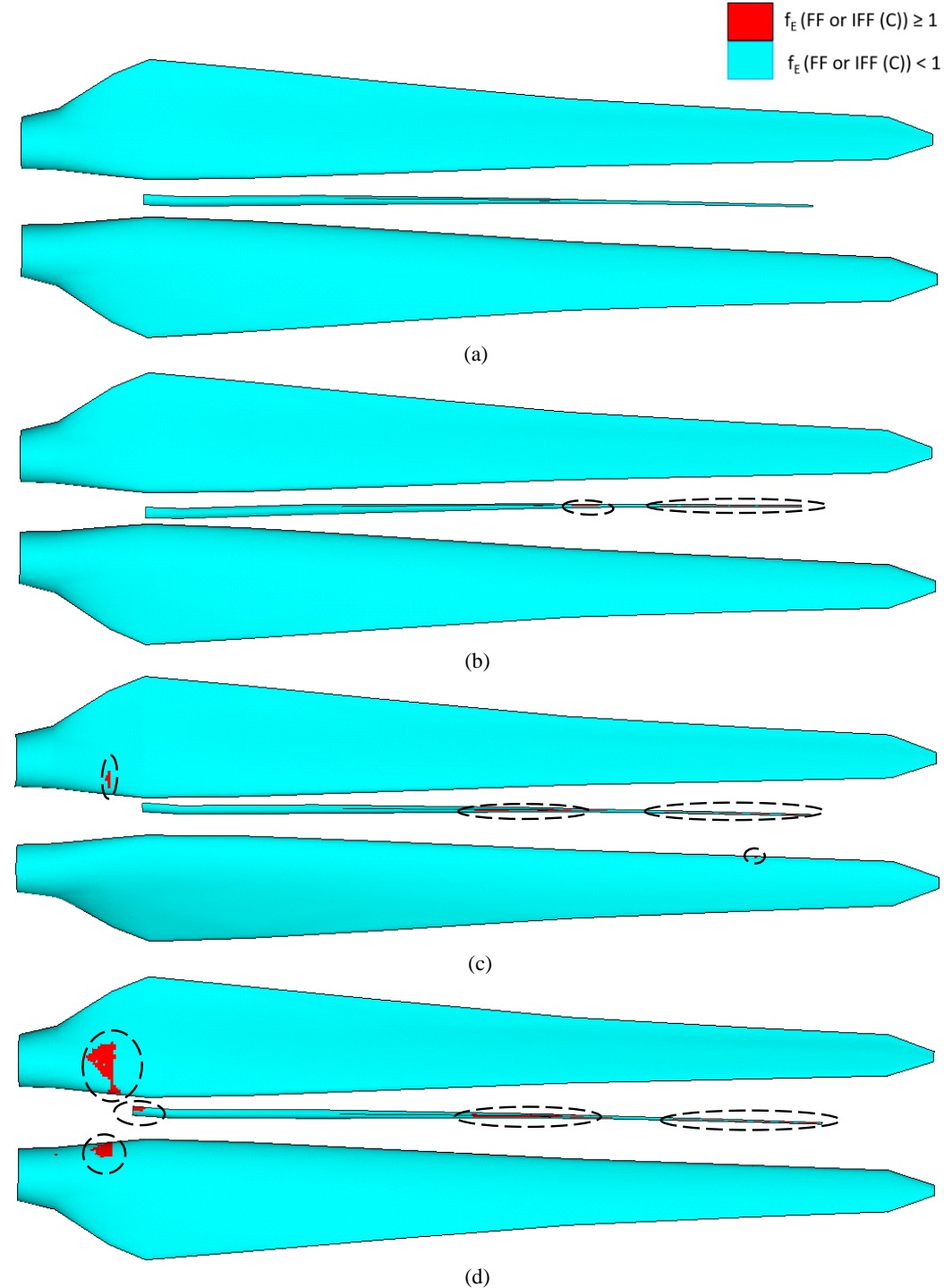

Figure 22. Element failure progression in the pressure side, internal flange and suction side of the blade (from top to bottom in a row) at (a) 100%, (b) 210%, (c) 309% and (d) %314 of extreme edgewise loading.

In Figure 23, inter-fiber failure mode A (IFF(A)) and inter-fiber failure mode B (IFF(B)) stress exposure distribution in pressure side, internal flange and suction side of the blade at 140% edgewise loading are shown. Based on the output data from the FE analysis of the blade, the damage initiation begins at 45% of extreme flap-wise loading due to IFF(A). Stress exposures greater than or equal to one indicate damage, and damaged regions are shown in red. The figure shows that inter-fiber failure begins in the internal flange of the blade. IFF(A) and IFF(B) do not lead to the element failure, but causes degradation in the transverse, shear moduli, and Poisson's ratio. The stress exposure at the leading edge near root and trailing are higher compared to other blade regions at this load level, which indicates damage growth at higher load levels in these areas. Under edgewise (min) loading condition leading edge close to blade root is subjected to compressive and trailing edge tensile stresses, respectively. Consequently IFF(A) caused by combination of tensile and shear stresses and IFF(B) caused by the combination of tensile and shear stresses is seen at the trailing edge and leading edge close to blade root, respectively. It is noted that FF and IFF(C) initiate also in the internal flange first but at higher load levels; 210% load. At 309% load FF and IFF(C) is seen at leading edge close to root, where compressive stresses are dominant. As the material has less strength under compression compared to tension, failure initiation begins at the leading edge close to root. These failures are in the same location as IFF(A) or IFF(B) which can be considered as subcritical cracks. As it was the case for flap-wise loading, this observation is similar to the statement (referring to Lambert et al. (2012) and Sorensen et al. (2004) studies) in Montesano et al. (2016) that subcritical cracks can act as precursor to more critical damage modes such as delamination or adhesive debonding.

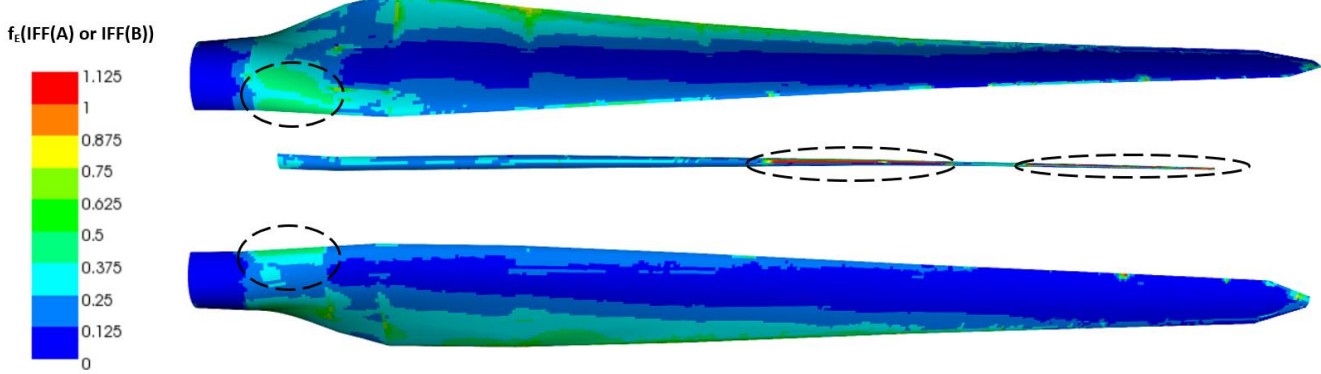

**Figure 23**. Damage evolution (IFF(A) or IFF(B)) in the pressure side, internal flange and suction side of the blade (from top to bottom in a row) at 140% of extreme edgewise loading.

### 3.2.3 Progressive damage analysis under combined edgewise (min) and flap-wise (max) loading

The total deformation of the nonlinear blade model versus the undeformed model under 100% combined extreme flap-wise and edgewise loading is displayed in **Figure 24**. The maximum blade deflection at the blade tip is 104 mm and less than the deformation in pure flap-wise loading. We further note that the deflection of 8 mm in the edgewise direction is much less than the deflection of 103 mm in the flap-wise direction.

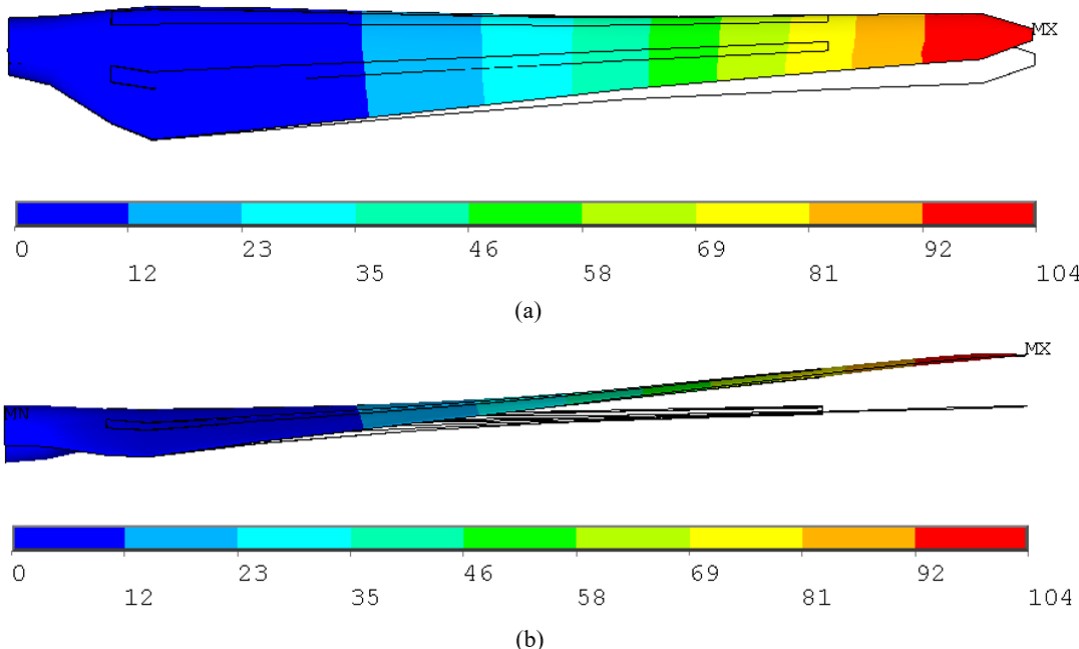

**Figure 24.** Total deformation of the RUZGEM blade model vs. undeformed model under 100% combined extreme edgewise and extreme flap-wise loading condition (a) top view (b) side view (scale factor: x2.5).

Load displacement curves in the range between 10% - 130% of combined extreme flap-wise and edgewise loading of the blade are displayed for the linear elastic model and progressive damage model (Puck) in Figure 25. In the figure, resultant edgewise 425 plus flap-wise loading versus resultant edgewise plus flap-wise loading is plotted. Element failure progression in the damaged blade components; pressure side, internal flange and suction side of the blade is depicted in Figure 27. The regions where the stress exposure IFF(C) or FF is equal to or greater than one is depicted in red. The stiffness of these elements is set to zero and these elements do not contribute to the blade strength anymore. In general, under combined loading element failure evolution is similar to pure flap-wise loading case, but failure occurs at higher load levels. Figure 25 shows that up to 85% extreme flap-430 wise loading, the stiffness for both linear elastic and progressive damage models remains almost the same. After 85% loading, slope reduction starts in the nonlinear progressive Puck model due to the element failures at the internal flange, suction, pressure side leading edges. This point is the first turning point in the load displacement curve. As the loading is increased to 100% element failure at the internal flange, suction, pressure side leading edges grows further. Second slope reduction is

detected at 118% loading. At this load level, in addition to the damaged region in the internal flange and leading edges, element
failure initiates at the trailing edge. This is a more obvious turning point compared to the first turning point at 85% loading. At
this load level Figure 27 (c) shows laminate failure in the internal flange, leading and trailing edges. Due to element failure,
deformation in the form of local buckling at the leading trailing edge is observed as depicted in Figure 26 below. In their study
regarding the full-scale testing of a 34-m wind turbine blade, under combined loading Haselbach and Branner (2016) also
observed laminate failure along the trailing edge. As the loading is further increased to 126%, element failure at the leading
and trailing edges covers a larger area. As a consequence, suction and pressure panels are detached and the blade integrity is
lost. Finally, after %126 loading blade is close to collapse. When the results in Figure 22 are compared with Figure 27, it can
be concluded that under same load level the number of failed elements under combined loading is less than pure flap-wise
loading case.

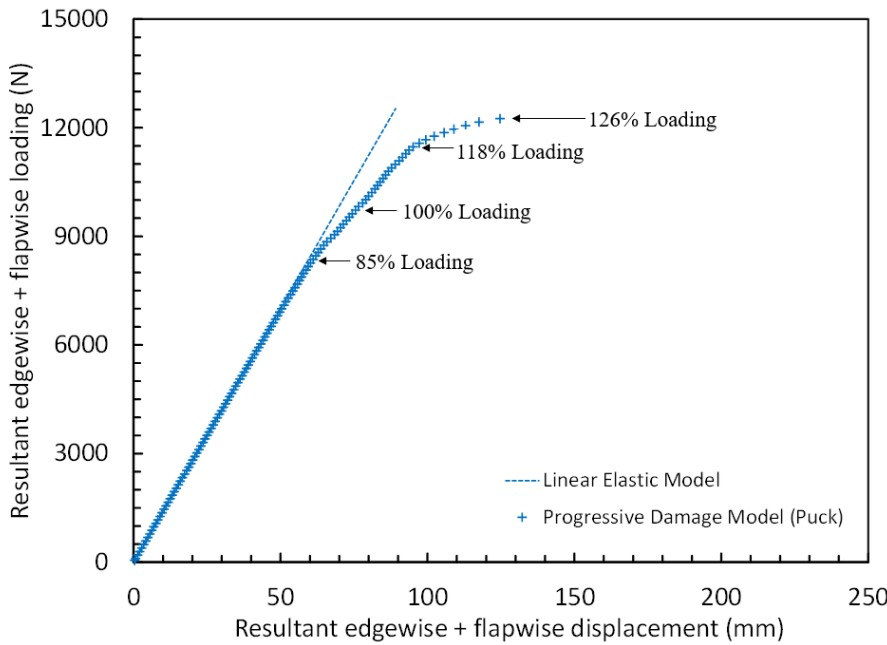

**Figure 25.** Load displacement curves of the blade using the linear elastic model and progressive damage model (Puck) under
under combined extreme edgewise (min) and extreme flap-wise (max) loading.

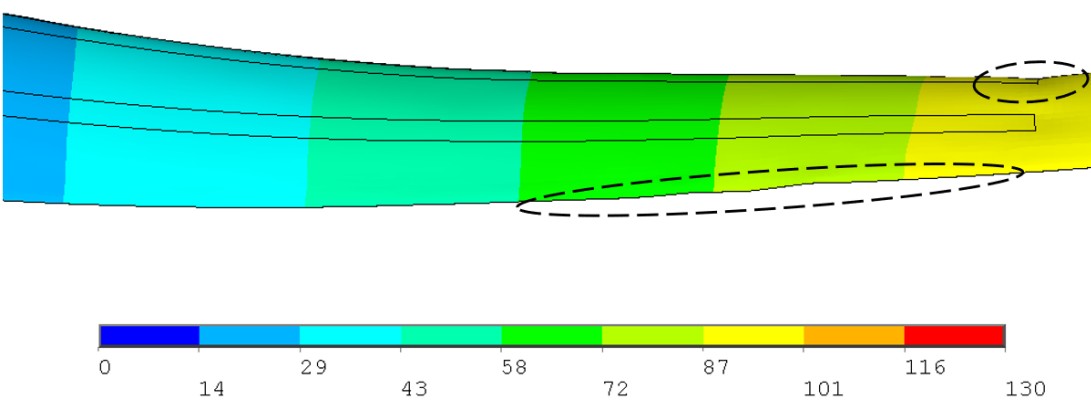

Figure 26. Total deformation occurring in the form of local deformation due to failed elements at 118% extreme combined edgewise and extreme flap-wise loading. (Scale factor for deformation plot: x15).

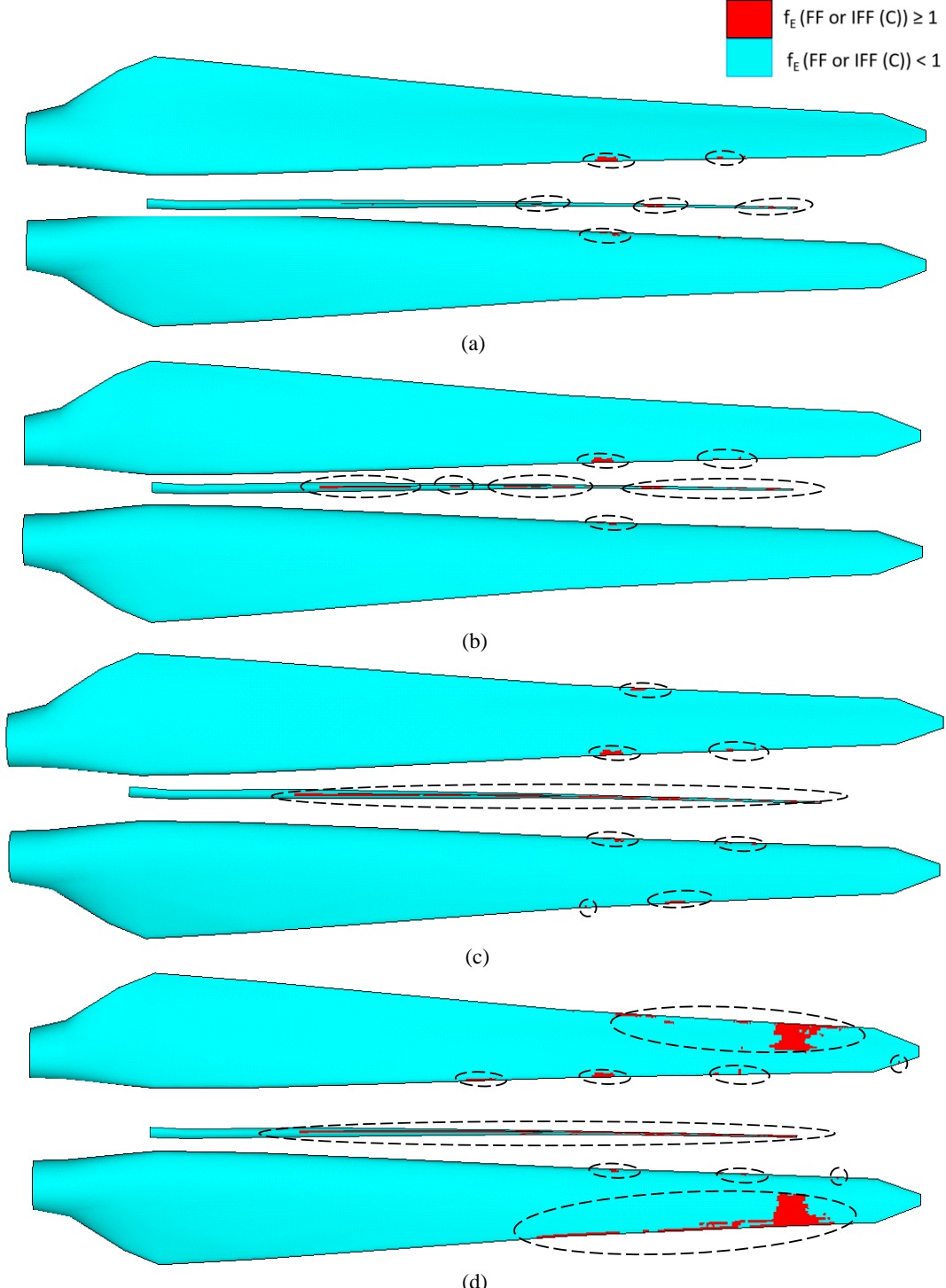

**Figure 27.** Element failure progression in the pressure side, internal flange and suction side of the blade (from top to bottom
in a row) at (a) 85%, (b) 100%, (c) 118% and (d) %126 of combined loading.

## 4    Discussion

In this section the progressive damage behavior of the blade under flap-wise (max), edgewise (min) and combined flap-wise (max) and edgewise loading (min) is discussed. From Figure 15, Figure 20 and Figure 25 it is seen that the slope of the load-displacement curve is highest for edgewise loading followed by edgewise plus flap-wise loading. For flap-wise loading, the slope of the load-displacement curve is the lowest. In Figure 28, instead of the resultant components of the combined loading, its flap-wise components, i.e., load and displacement in flap-wise direction, are plotted. It is seen that the blade exhibits a stiffer behavior in flap-wise direction under combined loading condition compared to pure flap-wise loading case. As a consequence, flap-wise deflection component under combined loading is approximately 85% of the flap-wise deflection under pure flap-wise loading. The blade is stronger under combined loading and at 116% loading and the damaged region is less under combined loading compared to flap-wise loading as shown in Figure 29.

In Figure 29, element failure progression is compared for edgewise, flap-wise, and combined loading scenarios under 116% loading. It is observed that the degree of the failed region is highest for the flap-wise loading case, followed by combined loading, and there is no failed region in the edgewise loading case. This failure development can be explained by the superposition of loads and stress components as tabulated in Table 3. For this study, points A, B and C are picked from the heavily damaged blade regions at 116% flap-wise loading as shown in Figure 29 (a). Since stresses cannot be read in regions where element failures are present, the study is carried out at 100% loading. The higher stress exposure fiber-failure (FF) or inter-fiber-failure mode C (IFF(C)), which cause laminate failure for the elements at points A, B and C are computed on a ply-by-ply basis. The critical ply with the highest stress exposure is detected and for this critical ply axial stress component ($\sigma_z$) is extracted. As seen from Table 3, at the selected points and the load cases, stress exposures for FF failure mode are dominant. Under flap-wise (max) loading case the critical plies are subjected to compressive axial stresses. When edgewise (min) loading is applied, tensile axial stresses are induced at the critical plies of the elements at points A, B and C. In the table, compressive stress components are written in red, which are seen under edgewise and combined loading. Under combined loading, compressive stress component due to flap-wise (max) and tensile stress component due to edgewise (min) loading are superimposed causing an overall reduction in the axial stress level as shown in Table 3. From the table, it is noted that the reduction in the axial stresses lead to a reduction of the stress exposures. It is also seen that the decrease in stress level is approximately 10-20%, which is in line with the 10% increase observed in the ultimate failure load value under combined loading compared to pure flap-wise loading. Consequently, the stress state in the blade caused by the superimposed stress components under combined loading decrease the extent of damage at the specified blade locations.

To summarize, finite element analysis of the RUZGEM 5-m blade using Puck's damage model indicates that laminate failure plays a major role for the ultimate blade failure. Laminate failure progression observed under flap-wise, edgewise and combined loading conditions fall into type 4 (internal damage formation and growth in laminates in skin) and type 5 (splitting and fracture of separate fibers in laminates of the skin) wind blade damages as categorized in Sorensen et al. (2014). Although

local buckling of small size wind turbine blades is found to be the major design concern in Paquette and Veers (2007) and Chen et al. (2015), the RUZGEM 5m blade is found to exhibit sufficient resistance against buckling in our investigation.

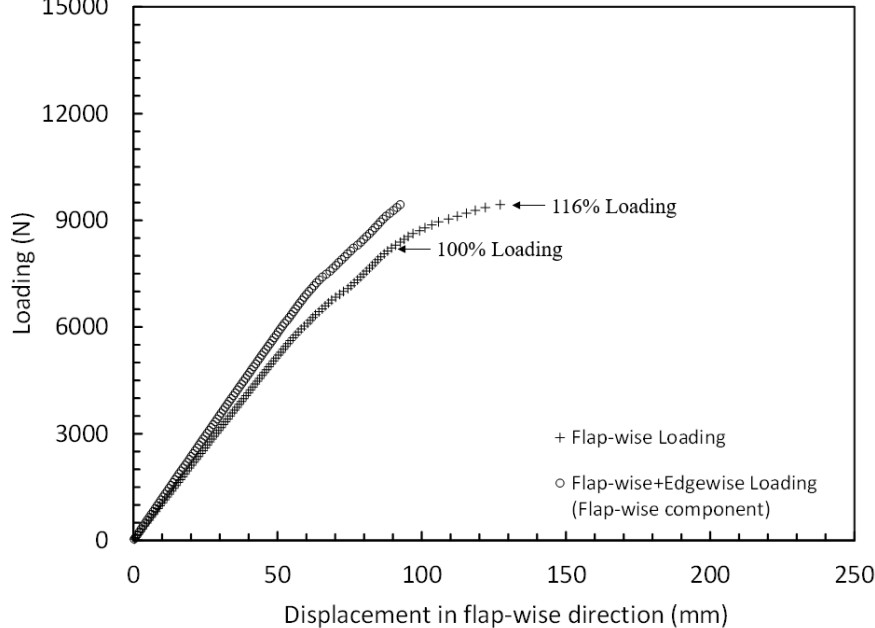

**Figure 28**. Load-displacement curves of the blade under pure flap-wise and combined loading using flap-wise components of the load and displacement.

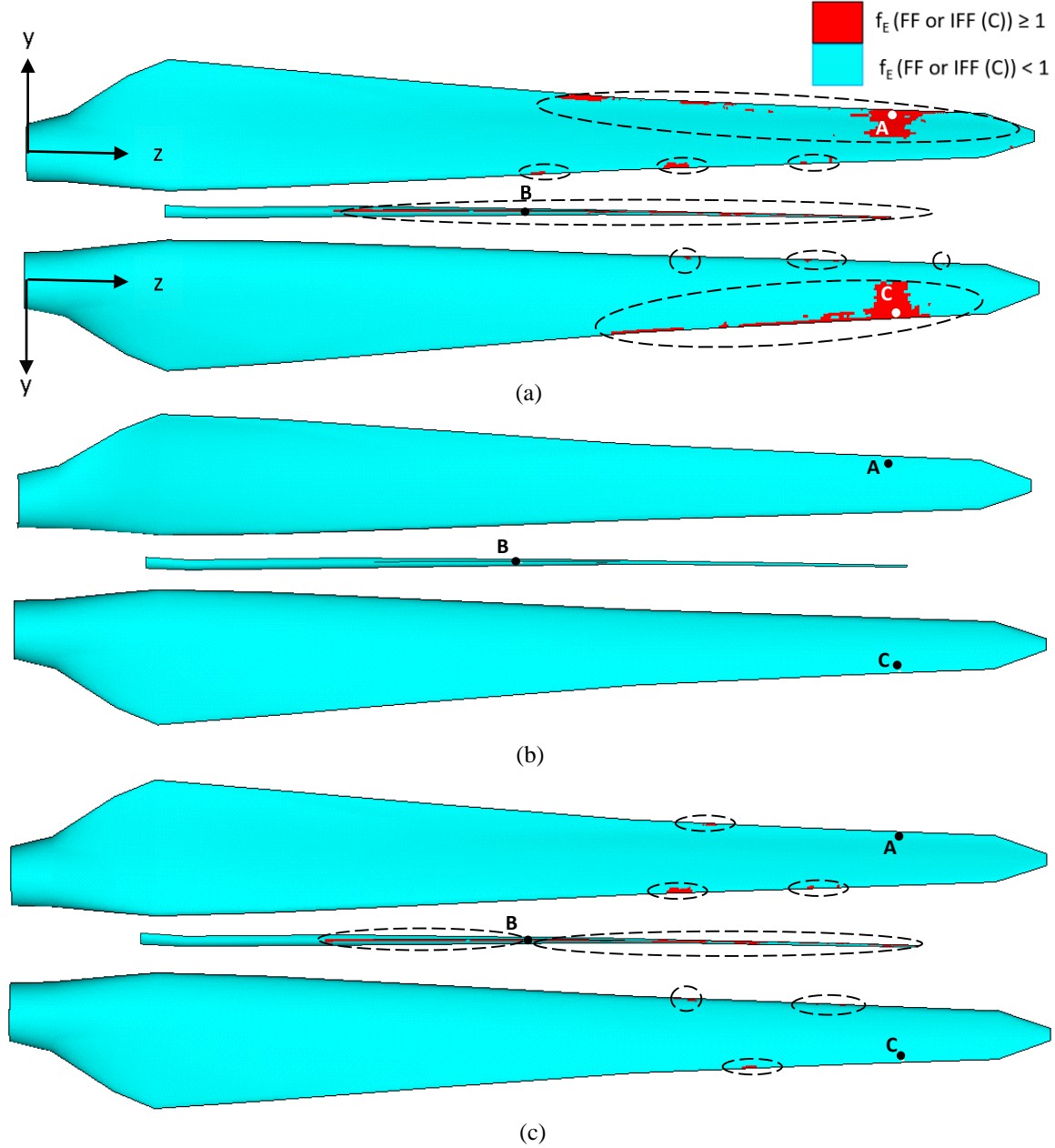

**Figure 29.** Element failure progression in the pressure side, internal flange and suction side of the blade (from top to bottom in a row) at (a) extreme flap-wise, (b) extreme edgewise (no element failure) and (c) combined extreme flap-wise and extreme edgewise at 116% extreme loading case. Points A (48,147,4301) at pressure side, B (-31,152,2224) at internal flange, C (55,176,4288) at suction side are used to interrogate the axial stress levels in the plies in Table 3 for 100% extreme loading.

**Table 3.** Axial stress levels and stress exposures in the critical regions of blade under flap-wise, edgewise and combined (flap-wise plus edgewise) at 100% extreme loading cases.

| Loading Condition | Pressure side (Point A) | | Internal Flange (Point B) | | Suction side (Point C) | |
|---|---|---|---|---|---|---|
| | Axial stress ($\sigma_z$) (MPa) | Stress exposure (FF) | Axial stress ($\sigma_z$) (MPa) | Stress exposure (FF) | Axial stress ($\sigma_z$) (MPa) | Stress exposure (FF) |
| Flap-wise(max) | -19.89 | 0.54 | -33.46 | 0.88 | -32.46 | 0.76 |
| Edgewise(min) | 0.98 | 0.01 | 2.13 | 0.08 | 1.86 | 0.02 |
| Combined | -17.97 | 0.50 | -25.19 | 0.68 | -28.31 | 0.69 |

## 5    Conclusions

In this work, strength characteristics of an existing 5-meter RUZGEM composite wind turbine blade under extreme flap-wise, edgewise and combined flap-wise and edgewise loading conditions are investigated. For this purpose, in addition to a linear buckling analysis, progressive damage analysis of the blade using Puck's (2D) physically based phenomenological model is performed. The main conclusions are as follows:

- Linear buckling analysis show that blade shows sufficient strength against buckling.

- Failure of elements due to IFF(C) or FF are observed, and a slope reduction in the load displacement is detected after the application of 75% extreme flap-wise loading and %85 combined loading cases. In contrast, under 100% edgewise loading element failures are not observed.

- For flap-wise and combined loading scenarios, a similar damage pattern is observed; laminate failure due to IFF(C) or FF in the internal flange causes the first slope reduction in load displacement curve. As the load is increased, damage grows along the trailing edge, which causes second slope reduction before collapse.

- For edgewise loading, laminate failure observed in the internal flange is the first slight slope reduction in load displacement curve. As the load is further increased, due to compressive stresses, damage accumulates at the leading edge close to blade root, which leads to second slope reduction before collapse.

- FF and IFF(C) initiate in the same location as IFF(A) or IFF(B). IFF(A) or IFF(B) denote subcritical ply cracks which precede more critical damage modes such as IFF(C) (which is an indicator of possible delamination), and FF.

- At the same load level, less damage is observed under combined loading compared to pure flap-wise loading. This damage evolution is attributed to the reduction of stresses (and thus stress exposure) caused by the superposition of stress components under flap-wise (max) and edgewise (min) loading conditions.

As a summary, trailing edge and internal flange located at the leading edge of the 5-m RUZGEM blade are found to be damaged primarily under flap-wise and combined loading conditions. For the edgewise loading, internal flange and leading edge close to blade root are the mainly damaged areas of the blade. It is noted that using the global modeling approach of the 5-m wind turbine blade with plane stress elements, through-the-thickness stresses, which are necessary for a detailed examination of delamination and/or debonding failures cannot be obtained. As a future work using sub-modeling technique with solid elements, competing failure mechanisms such as delamination and/or debonding can be investigated at the critical failure regions. As a follow-up study, full-scale structural tests for the existing 5-m RUZGEM wind turbine blade is planned following the completion of the testing facility. Afterwards, the structural response, primary damage zones and their development obtained from simulations will be compared with the experimental findings.

**Author contributions**. CM implemented the failure analysis method, conducted the numerical simulations, and wrote the
paper. DC is the supervisor and guided CM for the conception of the ideas and participated in writing, structuring, and
review of the paper.

**Data availability.** Blade geometry and layup that supports the results of this research is uploaded to the supplement.

**Competing interests.** The authors declare that they have no conflict of interest.

**Acknowledgement.** This work was partly supported by RUZGEM, METU Center for Wind Energy at the Middle East
Technical University.

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
