# Peer review of "Finite element simulations for investigating the strength characteristics of a 5-m composite wind turbine blade"

_Wind Energy Science, 2020_

## Author Comment (AC1) · 2 Apr 2020

**Virtual full-scale testing for investigating strength characteristics of a composite wind turbine blade**

Can Muyan[1,2], Demirkan Coker[1,2]

[1]Aerospace Department, Middle East Technical University, Ankara, 06800, Turkey
[2]Structural Mechanics and Materials Laboratory, RUZGEM (METUWIND) Center for Wind Energy Research, Middle East Technical University, Ankara, 06800, Turkey

*Correspondence to*: Demirkan Coker (coker@metu.edu.tr)

**Abstract.** Full-scale structural tests enable us to monitor the mechanical response of the blades under various loading scenarios. Yet these tests must be accompanied by numerical simulations so that the physical basis of the progressive damage development can be captured and interpreted correctly. Within the scope of this paper, the previous work of the authors concerning the strength analysis of an existing 5-m GFRP wind turbine blade using Puck failure criteria is revisited. A significant outcome of the previous study was that the nonlinear Puck material model is necessary for a more realistic simulation of failure mechanisms. In the current work, under extreme load cases, the internal flange at the leading edge and trailing edge of the blade are identified as the mainly damaged regions. Moreover, the dominant failure mechanism is expected to be the de-bonding at the trailing and leading edges. When extreme load case is applied as a combination of edge-wise and flap-wise loading cases, less damage is observed compared to the pure flap-wise loading case. This damage evolution is attributed to the stiffer structural behavior of the blade under combined loading condition.

**1 Introduction**

As fundamental eco-friendly renewable energy resources, wind turbines are designed to operate over a lifespan of 20 years. According to Holmes et al. (2007), long-term structural reliability of wind turbine components is vital when the high cost of manufacturing, inspection, and repair, especially for turbines located in remote regions, are considered. Composite blades are among the most critical components of a wind turbine, which are subjected to complex loading conditions. A rotor blade failure can have a significant impact on turbine downtime and safety. In order to assure sufficient mechanical resistance, structural testing and analysis must be conducted. However, structural testing methods such as full-scale testing of the blade are expensive and troublesome due to the construction of a test set-up. In order to capture and understand the physical basis of the progressive damage development correctly, tests need to be accompanied by numerical analysis methods (Chen et al., 2017). Moreover, structural analyses are utilized to calibrate structural blade test set-ups for different loading conditions. In the literature, there are many studies on the structural behavior of composite turbine blades. A novel methodology for the structural design and analysis of 8-m tidal current turbine blade is presented based on the Puck phenomenological failure criteria for fibre and inter-fibre failure by Fagan et al. (2016). The methodology that is developed in the study predicted

damage values for different load cases. This methodology is an iterative design process using the failure criteria to check the structural strength of the blade. Their turbine blade Finite Element Model did not include geometric nonlinearity. Passipoularidis et al. (2011) developed a fatigue damage simulator (FADAS) utilizing Puck failure criteria for the life prediction of GFRP laminates, which are commonly used for the construction of wind turbine blades under variable amplitude loading. In Passipoularidis et al. (2011)'s study, failure analysis is done on the ply level based on Classical Lamination Theory (CLT). Puck failure criteria are implemented to predict failure initiation and sudden stiffness degradation. The predictions of FADAS agree well with fatigue data and show that the algorithm is able to take into account load sequence effects, as well. Another study conducted by Jensen et al. (2006) is about the full-scale test and nonlinear FEM simulation of a 34-meter composite wind turbine blade under flap-wise loading. Load-displacement curves are used to predict the location of failure initiation, which leads to ultimate failure. After comparing the test with simulation, delamination of the skin and the following buckling was found to be the main failure mechanism for ultimate collapse. Chen et al. (2017) revisited their 
[revised manuscript text omitted]
. This expectation can be justified by the variation of the contact friction stress in trailing edge adhesive interface as depicted in Figure 13. At 100% flap-wise loading the evolution of the failed elements is shown in black and denoted by region A in Figure 12 (b). At the same location we observe a significant increase in contact friction stress as denoted by region A in Figure 13. This contact stress peak in region A can lead to de-bonding. Furthermore, bearing in mind that adhesive is modeled as bonded contact, the current modeling technique is not capable of simulating the progression of debonding. It is worth noting that, plane stress elements, which are used to model the blade are not able to show through-the-thickness stresses, which may futher help to trigger debonding/delamination failure mechanisms.

[Figure]

**Figure 12.** Element failure progression on the suction side of the blade at (a) 90%, (b) 100%, (c) 120% and (d) %150 of extreme flap-wise loading.

[Figure]

**Figure 13.** Contact friction stress in the trailing edge adhesive interface along blade spanwise direction at 100% extreme flap-wise loading.

In Figure 14 inter-fiber failure mode A (IFF(A)) and inter-fiber failure mode B (IFF(B)) distribution in the suction side are shown on the same plot. If both failure exposures are present in an element, the higher failure exposure IFF(A) or IFF(B) is shown. Based on the output data from the virtual full-scale test of the blade, the damage initiation begins at 40% of extreme flap-wise loading due to IFF(A) or IFF(B). The figure shows that inter-fiber failure starts at the trailing edge of the suction side towards spar end. As discussed in the methodology chapter, failure exposures greater than or equal to one indicate damage, and damaged regions are shown in red. Under 40% to 100% loading damage propagates by increasing load increments, as shown in Figure 15. From the regions where inter-fiber failure values are high, detail section D is created on the blade suction side. This detailed section is used to study the damage evolution in Figure 15. After the beginning of failure at 40%, the damaged regions grow along the trailing edge towards the blade tip and root. It is worth noting that, IFF(A) and IFF(B) do not lead to the element failure. When IFF(A) or IFF(B) occurs, only the transverse, shear moduli, and Poisson's ratio are reduced according to the degradation rules. As seen from Figure 10, degradation in the stiffness and Poisson's ratio does not lead to the change in the slope of the force-displacement curve. This observation means there are no considerable changes in the blade stiffness. However, at 100% load, there are failed elements due to FF or IFF(C), as will be discussed in the preceding paragraphs, and at this point, the slope of the force-displacement curve, i.e., blade stiffness decreases. Since the IFF(A) or IFF(B) of the failed elements are zero, they appear as dark blue regions inside red areas. When the elements fail, they do not contribute to strength anymore, and the load will be carried by the neighboring elements. As a result, IFF(A) or IFF(B) evolves around the failed elements.

[Figure]

**Figure 14.** Detail section D from the suction side of the blade at 40% of extreme flap-wise loading.

[Figure]

**Figure 15.** Failure progression on blade suction side at (a) 40% and (b) 100% of extreme flap-wise loading in detail D.

265     Damage progression regarding IFF(C) and FF in detail section D of the blade is further investigated in Figure 16. In Figure

16, fiber-failure (FF) and inter-fiber failure mode C (IFF(C)) distribution in the suction side is shown on the same plot. If

both failure exposures are present in an element, the higher failure exposure FF or IFF(C) is shown. In the figure, failure

exposures are plotted under 90%, 100%. 120% and 150% loading. At 90% loading, the regions depicted in red are the

regions where the failure index is equal to or greater than one, the stiffness of the elements is set to zero. As a consequence,

270     in the following load increment of 100% loading, the regions, which appear in red at 90% loading, become dark blue. After

setting their stiffness to zero, their failure exposure FF or IFF(C) is zero, and they do not contribute to the strength of the

blade anymore. We note that these dark blue regions correspond to the failed elements in Figure 12. At 100% loading, the

failed region evolves along the trailing edge, as depicted in red. The new failed regions, together with the failed elements

from previous runs, are represented by the failed elements in black in Figure 12. We note FF and IFF(C) initiate in the same

275     location as IFF(A) or IFF(B).

[Figure]

**Figure 16.** Failure progression on the blade suction side at (a) 90%, (b) 100%, (c) 120% and (d) %150 of extreme flap-wise loading in detail D.

**3.2 Virtual full-scale testing under edgewise loading**

Load displacement curves in the range between 10% - 350% of extreme flap-wise loading of the blade are displayed for the linear Puck model and nonlinear progressive Puck model in Figure 17. Loads are computed from the reaction forces at the blade root, and the displacement is measured at a node near the blade tip. It can be observed from the figure that up to 320% of the extreme flap-wise loading, the stiffness for both models remains almost the same. After 320% loading, stiffness reduction starts in the nonlinear progressive Puck model due to the first element failure at the internal flange tip. Finally, at %350 loading blade is close to collapse.

[Figure]

**Figure 17.** Load displacement curves of the blade using the linear Puck model and nonlinear Puck progressive model under edgewise loading.

The total deformation of the nonlinear blade model versus the undeformed model under 100% extreme flap-wise loading is displayed in Figure 18. The maximum blade deflection at the blade tip is 27 mm and much less than the deformation compared to pure flap-wise loading. Figure 19 shows the evolution of the failed elements due to FF and IFF(C) in the suction side of the blade at 320%, 330%, 340% and 350% of extreme edgewise loading. Element failure in the trailing edge near root begins at 330% loading, and the failed elements are shown in black. We observe from Figure 19 that since the spar has more structural strength than the suction side in edgewise direction, damage grows inside the suction side further and ends at the boundary where the spar is located until %340 loading. As seen from the figure number of failed elements increase in the trailing edge towards the spar and root as the load is further increased to 350%. Starting from 350% loading, the part of the suction side connected to the spar caps is heavily damaged.

[Figure]

**Figure 18.** Total deformation of the nonlinear METUWIND blade model vs. undeformed model under 100% extreme edge-
300   wise loading condition (scale factor: x10).

[Figure]

**Figure 19.** Element failure progression on the suction side of the blade at (a) 320% (no element failure) (b) 330%, (c) 340%
and (d) 350% of extreme edgewise loading.

In Figure 20, fiber-failure (FF) and inter-fiber failure mode C (IFF(C)) distribution in the suction side are shown on the same plot. In the figure, failure exposures are plotted under 320%, 330%, 340%, and 350% loading. The regions where the failure index is equal to or greater than one are depicted in red. The stiffness of these elements is set to zero. These regions, which appear as dark blue regions in Figure 20 (c) and (d) correspond to the failed elements in Figure 19. At 340% and %350 loading cases, the failed region evolves along the trailing edge, as depicted in red. The new failed areas, together with the failed elements from previous runs, are represented by the failed elements in black in Figure 19. Based on the results, blade design exhibits excessive safety in edgewise direction and is considered to be over-conservative for this type of loading.

[Figure]

**Figure 20.** Failure progression on the blade suction side at (a) 320%, (b) 330%, (c) 340% and (d) 350% of extreme edgewise loading.

**3.3 Virtual full-scale testing under combined edgewise and flap-wise loading**

Load displacement curves in the range between 10% - 160% of combined extreme flap-wise and edgewise loading of the blade are displayed for the linear Puck model and nonlinear progressive Puck model in Figure 21. Loads are computed from the reaction forces at the blade root, and the displacement is measured at a node near the blade tip. The figure shows that up to 110% extreme flap-wise loading, the stiffness for both models remains almost the same. After 110% loading, stiffness reduction starts in the nonlinear progressive Puck model due to the first element failure at the internal flange tip. Finally, at %160 loading blade is close to collapse.

[Figure]

**Figure 21.** Load displacement curves of the blade using the linear Puck model and nonlinear Puck progressive model under combined edgewise and flap-wise loading.

325     The total deformation of the nonlinear blade model versus the undeformed model under 100% combined extreme flap-wise and edgewise loading is displayed in Figure 22. The maximum blade deflection at the blade tip is 103 mm and less than the deformation in pure flap-wise loading. We further note that the deflection in the edgewise direction is less than the deflection in the span-wise direction. Figure 23 shows the evolution of the failed elements due to FF and IFF(C) in the suction side of the blade at 100%, 110%, 120%, and 160% of the combined extreme edgewise and flap-wise loading. As in the pure flap-

330     wise loading case, element failure distribution is represented in black. Yet under the same load increment, the number of failed elements are less compared to pure flap-wise loading case. This observation is due to resistance, i.e., stiffer behavior of the blade to bending under combined loading case. Element failure in the trailing edge begins at 110% loading. Similar to the pure flap-wise loading, the number of failed elements increases in the trailing edge towards the spar and root as the load is further increased up to 160% loading. Since compared to edgewise loading the spar stiffness in the flap-wise direction is

335     less, at 160% combined loading damage grows further inside the suction side where the spar is located.

[Figure]

(a)

(b)

[revised manuscript text omitted]

**4 Conclusions**

385 In this work, virtual full-scale test of an existing 5-meter composite wind turbine blade under extreme flap-wise, edgewise and combined flap-wise and edgewise loading conditions is conducted using Puck material damage model. Failure on the lamina level is investigated using Puck failure criteria (2D). The main conclusions are as follows:

1. Failure of elements are observed, and the blade is found to deflect extensively after the application of 90% extreme
390 flap-wise loading, ultimate failure is expected to occur. In contrast, element failures are not observed under pure edgewise and combined edgewise and flap-wise extreme load cases until 100% extreme loading.

[revised manuscript text omitted]

---

## Referee Comment (RC1) · Xiao Chen (Referee) · 9 Apr 2020

1. In essence, this work presented FE simulations of a 5-m full-scale blade subject to static loads using nonlinear puck failure criterion. Instead of using 'virtual full-scale testing', it is more suitable to use 'FE simulations' to reflect the essence of this work. To the reviewer, 'virtual testing' is more than FE simulations. 2. In the abstract, 'so that the physical basis of the progressive damage development can be captured and interpreted correctly.' should change to 'so that the physical basis of the progressive damage development can be better interpreted and understood'. Physical tests capture real damages while FE simulations hopefully can complement experimental observations to achieve better understanding. You may consider to read https://doi.org/10.1016/j.compstruct.2019.03.018

3. The big blades behave/fail differently from the smaller ones. Two studies listed below showed that the governing failure mechanisms are quite different. How the results from this study using a 5-m blade FE model are relevant to the blades which are usually more than 10 times longer? Please comment on this. https://doi.org/10.1007/s11431-014-5741-8 and https://doi.org/10.3390/en7042274

4. In Chen et al. (2017), 3D stresses/strains are found to be important in the failure of a 52.3m blade and solid elements are recommended in FE simulation when the failure is of concern. Please comment on the shell elements used in this study and maybe state the scope of this study in the introduction.

5. In Fig. 4, it seems that the stress-strain curves are rather linear. How can one see that the nonlinear Puck material damage model used in this study is superior to other models, even to the linear ones?

6. In Fig. 4, please also show the comparison when the other models are used, e.g., the normal Puck, Tsai-Wu, etc.

7. In Fig. 10 and the other similar figures, please compare when the other models are used, e.g., the normal Puck, Tsai-Wu, etc. Modern FE software provide the built-in composite damage models for shell elements, please include the comparison in relevant curves.

8. In Fig. 12 (d), why there is considerable damage at the blade tip, which is usually not loaded.

9. in Fig. 15(d), why there is an undamaged region (blue) enclosed by the damaged region (red)?

10. In Fig. 16 and other similar figures, please also show what is happening in the blade model at the turning points, is it due to local or global buckling?

11. In Fig. 19(c) and 19(d), why do the damaged regions heal? It is better to show the damage status rather than the Puck index. Like the one used in https://doi.org/10.3390/en7042274

---

## Referee Comment (RC2) · Xiao Chen (Referee) · 7 May 2020

The authors have addressed all the comments from the reviewer and improved the manuscript to a publishable level. It is suggested to accept this manuscript for publication in Wind Energy Science. Congratulations to the authors! (One minor editorial modification is Fig. R12. The scale of the horizontal axis could be reduced, e.g, to 100 mm, for better readability.)
* * *

---

## Author Comment (AC2) · 7 May 2020

We thank the Referee for the detailed and insightful comments regarding the paper. In our response, we have carried out new simulations in light of his comments where necessary and have added new figures that correct and clarify our work. The comments are individually addressed below, with the Referee's comments written in red and our response in black. If accepted, our response to the Referee's comments will be incorporated in the revised version of the paper which we believe will result in a clearer and more thorough paper.

**Comment 1:** In essence, this work presented FE simulations of a 5-m full-scale blade subject to static loads using nonlinear puck failure criterion. Instead of using 'virtual full-scale testing', it is more suitable to use 'FE simulations' to reflect the essence of this work. To the reviewer, 'virtual testing' is more than FE simulations.

**Response to Comment 1 :**
We agree with the referee and we propose to change the title to:
"FE simulations to investigate the strength characteristics of a 5-m composite wind turbine blade" where we also limited the scope to the existing 5-m blade that we are investigating.

**Comment 2:** In the abstract, 'so that the physical basis of the progressive damage development can be captured and interpreted correctly.' should change to 'so that the physical basis of the progressive damage development can be better interpreted and understood'. Physical tests capture real damages while FE simulations hopefully can complement experimental observations to achieve better understanding. You may consider to read https://doi.org/10.1016/j.compstruct.2019 .03.018.

**Response to Comment 2 :**
We agree with the Referee and the abstract is changed according to the comment. In addition, wording used in the abstract and other parts of the paper is replaced with a more precise wording to accurately reflect what we mean.

**Comment 3:** The big blades behave/fail differently from the smaller ones. Two studies listed below shows that the governing failure mechanisms are quite different. How the results from this study using a 5-m blade FE model are relevant to the blades which are usually more than 10 times longer? Please comment on this.
https://doi.org/10.1007/s11431-014-5741-8 and https://doi.org/10.3390/en7042274

**Response to Comment 3:**
Chen, Wei Zhao, Lu Zhao and Xu (2014) conducted a full-scale bending test of a 52.3 m wind turbine blade and found that delamination in the spar cap and shear web failure at the root transition region were the main failure mechanisms for the blade collapse. Local buckling contributed to the main failure mechanism by facilitating local out-of-plane deformation. They conclude that for large blades, through-the-thickness stresses which cause debonding and delamination at the blade root transition region need to be considered in the FEA. Chen, Qin, Yang, Zhao and Xu (2015) focused on the local buckling resistance of 10.3 m wind turbine blades. FE analysis showed that configurations with sharp edges are susceptible to local buckling. During testing of the 10.3 m blade, although local buckling of shear web and flatback airfoil was observed, composite laminate failure in these locations was not observed. These results indicate different mechanisms for different blade sizes.

Within the framework of the current study, strength of an existing 5m blade is studied in terms of composite material failure. Further, in response to Referee's comment 10, a linear buckling analysis is conducted. The 5m blade is found to exhibit sufficient resistance against buckling in our investigation (see response to comment 10). However, our current model using Puck's damage model indicates that laminate failure plays a major role for the ultimate blade failure. Our analysis results suggest that debonding and delamination analysis can be useful to properly interpret the results. In addition to damage in the trailing edge, we found that the thin and stiff internal flange located at the leading edge is damaged primarily under flap-wise loading condition for the 5m blade. The simulation results will be compared with experiments to be conducted at RUZGEM on a 5 m blade which will be a follow-up to this study.

**Comment 4:** In Chen et al. (2017), 3D stresses/strains are found to be important in the failure of a 52.3m blade and solid elements are recommended in FE simulation when the failure is of concern. Please comment on the shell elements used in this study and maybe state the scope of this study in the introduction.

**Response to Comment 4:**

The scope of our investigation is limited to the finite element investigation of the strength of an existing 5m blade in terms of composite material failure.

Following the suggestion of the Referee, the following paragraph will be included in the introduction of the manuscript:

"Chen, Zhao and Xu (2017) (https://doi.org/10.1002/we.2087) found 3D stresses/strains to be important in the failure of a 52.3-meter blade and solid elements recommended in the FE simulation when debonding failure is of concern. They utilized 3-D strains and Yeh-Stratton failure criterion to calculate delamination and debonding failures in the blade. The scope of this work is limited to the investigation of the structural response of a 5-meter blade using global Finite Element Modeling approach and progressive composite failure analysis. For this structural analysis global FE Model is meshed using plane stress shell elements. Using the current modeling technique with shell elements critical locations for failure and worst load case scenario are identified. Puck's 2-D damage model demonstrates the direction to proceed for a complete and comprehensive modeling of the failure mechanisms. Furthermore, within the framework of this study, differences between edgewise, flap-wise and combined flap-wise/edgewise loading conditions are discussed."

**Comment 5:** In Fig. 4, it seems that the stress-strain curves are rather linear. How can one see that the nonlinear Puck material damage model used in this study is superior to other models, even to the linear ones?

**Response to Comment 5:**
Following the comment of the Referee, we notice that we misused terms in our paper that led to confusion and misunderstanding. We made changes to the terminology. We changed the term "linear model (Puck)" to the correct term "linear elastic model" and "nonlinear model (Puck progressive)" to "progressive damage model". This is more appropriate since in the "linear model (Puck)", only linear elastic model is used and no damage is implemented. In the "nonlinear model", progressive damage algorithm is used. The text will be corrected with correct terminology in the revised manuscript. As stated in the methodology section of the manuscript, Puck's progressive damage model (Puck and Schuermann, 1998) (https://doi.org/10.1016/S0266-3538(96)00140-6) is implemented in the ANSYS. Puck's model (Puck and Mannigel, 2007) (https://doi.org/10.1016/j.compscitech.2006.10.008) which incorporates "non-linear stress-strain relations for the inter-fibre fracture analysis of FRP laminates" is not used in this study.

Regarding our decision for the damage model used in our study, we would like to point out that all failure criteria have advantages and disadvantages. We decided to use Puck's progressive damage model because Puck's theory is based on fracture planes and Mohr-Coulomb's hypothesis and enables differentiation between fiber and three different inter-fiber failure modes (IFF(A), IFF(B) and IFF(C)). We have chosen to implement Puck's failure model to have a more detailed understanding of the failure modes. In the past, the authors have compared Puck's and Tsai-Wu failure models, albeit nonprogressive implementation, for the strength analysis of a 5-m wind turbine blade and found that Tsai-Wu (1971) delivers more conservative results compared to Puck (Ozyildiz, Muyan, Coker, IOP TORQUE 2018) (https://doi.org/10.1088/1742-6596/1037/4/042027).

Comparison of Puck with Tsai-Wu and other failure criteria was not studied in the manuscript because it was not in the scope of this study. However, comparison was made in the literature, namely World Wide Failure Exercise (WWFE) I Part B (Hinton et al, 2002) (https://doi.org/10.1016/S0266-3538(02)00125-2) for the original Tsai- Wu (1971) (https://doi.org/10.1177%2F002199837100500106) which did not perform as well as Puck (1998). As an example, comparison between the predicted and measured biaxial failure envelopes for [0/±45/90]s CFRP/AS4 3501-6 laminates shown in Figure R 1 demonstrates that in the third quadrant under compression-compression stresses Tsai (1971)'s prediction becomes quite non-conservative compared to the prediction of other failure theories. Moreover, the original Tsai-Wu (1971) is not capable of differentiation between different failure modes whereas DNV GL Standard (2015) (https://rules.dnvgl.com/docs/pdf/DNVGL/ST/2015-12/DNVGL-ST-0376.pdf) requires a separate strength verification for fiber and inter-fiber failure modes such as Puck or LARC03.

[Figure]

(a)                   (b)

Figure R 1. (a) and (b) Comparison between the predicted and measured biaxial failure envelopes for [0/±45/90]s CFRP/AS4 3501-6 laminates (Hinton et al., 2002) (https://doi.org/10.1016/S0266-3538(02)00125-2).

**Comment 6:** In Fig. 4, please also show the comparison when the other models are used, e.g., the normal Puck, Tsai-Wu, etc.

**Response to Comment 6:**
As mentioned in answer 5, we changed the misleading terminology from linear model (Puck) to linear elastic model and nonlinear to progressive damage model. The text will be corrected with correct terminology in the revised manuscript.

The comparison recommended by the Referee has been carried out in the literature. Figure R 1 shows Fig. 4 as taken from the World Wide Failure Exercise (WWFE) I Part B (Hinton et al, 2002) that compares the stress-strain curves for [0/90]s GFRP and [0/±45/90]s CFRP laminates. As seen from Figure R 2, Tsai-Wu (1971) underpredicted the laminate strength for GFRP/MY750 and CFRP/AS4 3501-6 laminates under uniaxial tension. If the Referee recommends the comparison of the modified version of the original Tsai-Wu (1971) model as explained in Chen, Wei Zhao, Lu Zhao and Xu (2014) in which failure mode-based material degradation is used, it can be implemented in the revised manuscript.

We would like to emphasize that these comparisons are for the original Tsai-Wu (1971) while the modified version in the study (Liu and Tsai, 1998) (https://doi.org/10.1016/S0266-3538(96)00141-8) has been shown to be more accurate as stated in the conclusions: "the Tsai theory proved to be in the leading group of those tested in the 'exercise'."

[Figure]

Figure R 2. (a) Comparison between measured and predicted stress-strain curves for [0/90]s GFRP/MY750 laminate under uniaxial tension $\sigma_y = 0$ (b) Comparison between measured and predicted stress-strain curves for [0/±45/90]s CFRP/AS4 3501-6 laminate under uniaxial tension $\sigma_x = 0$ (Hinton et al., 2002) (https://doi.org/10.1016/S0266-3538(02)00125-2).

**Comment 7:** In Fig. 10 and the other similar figures, please compare when the other models are used, e.g., the normal Puck, Tsai-Wu, etc. Modern FE software provide the built-in composite damage models for shell elements, please include the comparison in relevant curves.

**Response to Comment 7:**

As mentioned in response 5, we changed the terminology linear model (Puck) to linear elastic model and nonlinear model (progressive Puck) to progressive damage model. The text will be corrected with correct terminology in the revised manuscript.

We are using ANSYS Version 17.2 for our FE Simulations and only the original version of Tsai-Wu (1971) model is available as the built-in Tsai-Wu composite damage model in ANSYS. Since Tsai-Wu (1971) did not perform as well as Puck (1998) failure criteria as detailed in our response to comments 5 and 6, we did not compare Puck with Tsai-Wu. If recommended by the Referee, a new ANSYS APDL script can be written in order implement the modified Tsai-Wu model explained in Chen, Wei Zhao, Lu Zhao and Xu (2014).   Afterwards, the comparison can be done. However, we believe comparison of advanced failure models is outside the framework of this paper.

**Comment 8:** In Fig. 12 (d), why there is considerable damage at the blade tip, which is usually not loaded.

**Response to Comment 8:**
We like to thank the referee for this observation, and as a result we conducted a detailed investigation for the damage at the blade tip. We noticed unmerged nodes at the suction side tip and created a new finer mesh as seen in Figure R 3 below. In addition to the merging nodes at the tip of the suction side, the overall mesh structure of the blade is improved. In the updated FE Model of the blade there are 63104 elements. After running the updated FE Model Fig. 10 (Figure R 4) and Fig. 12(Figure R 5) in the manuscript are updated as seen below.

Element failure progression in the pressure side, internal flange and suction side of the blade is depicted in Figure R 5 below. According to the analysis results, element failure is observed in the internal flange at 90% loading. As seen from Figure R 5 failure in the internal flange grows further as the load is increased to 100%. At 160% loading in addition to the damaged region in the internal flange, damage grows along the trailing edge, pressure side and leading edge. Shortly before collapse at 180% loading, damaged regions at the leading and trailing edges evolve further and damage at the blade tip occurs. The reasons for damage initiation at the blade tip at the most extreme load level can be explained as follows:

- From Figure 7 in the manuscript it is seen that there is although low, some loading on the blade tip. At 180%, 1.8 times the extreme flap-wise loading, which read is from Figure 7 is applied to the blade and the blade collapses afterwards.

- Blade tip structure is rather thin and less stiff compared to other regions of the blade.

- As seen in Figure R 5(d) at 180% load level trailing edge and the internal flange which is used to bond the pressure and suction sides of the blade are already damaged. As a consequence, towards the blade tip the pressure and suction sides of the blade are detached at this load level. Under these circumstances blade tip structure is weaker and can be damaged more easily.

[Figure]

Figure R 3. Correction of the unmerged nodes at the blade suction side tip.

[Figure]

Figure R 4. Load displacement curves of the blade using the linear elastic model and progressive damage model (Puck) under extreme flap-wise loading. (This figure updates Figure 10 in the original manuscript).

[Figure]

Figure R 5. Element failure progression in the pressure side, internal flange and suction side of the blade (from top to bottom in a row) at (a) 90%, (b) 100%, (c) 160% and (d) %180 of extreme flap-wise loading. (This figure updates Figure 12 in the original manuscript).

**Comment 9:** in Fig. 15(d), why there is an undamaged region (blue) enclosed by the damaged region (red)?

**Response to Comment 9:**
After running the updated FE Model (See response 8) Fig. 15 (Figure R 6) in the manuscript are updated as seen below.

Damage progression regarding inter-fiber-failure (IFF) C and Fiber-failure (FF) in detail section D of the blade is investigated in Figure R 6. IFF (C) and FF stress exposures (Puck's terminology for failure index) are shown on the same plot. If both IFF (C) and FF failures are present in an element the greater stress exposure FF or IFF (C) is depicted.

In Figure R 6 (d) dark blue regions enclosed by the damaged region (red) are the 'killed' elements from previous load steps. At the end of a load step, if the stress exposure FF or IFF(C) exceeds one (red regions), the elements are deactivated by the EKILL command in ANSYS. A deactivated element remains in the model but contributes an almost zero value to the overall stiffness matrix. Dark blue represents 'zero' stress exposure, because there exists no stress in these 'killed' elements. The new damaged region in red evolves around the 'killed' elements. The new damaged region in red together with dark blue region form the new group of 'killed' elements. This new group of 'killed' elements are the failed elements depicted in red in Figure R 5 (d).

[Figure]

Figure R 6. Failure progression on the blade suction side at (a) 90%, (b) 100%, (c) 160%, and (d) %180 of extreme flap-wise loading in detail D. (This figure updates Figure 15 in the original manuscript).

**Comment 10:** In Fig. 16 and other similar figures, please also show what is happening in the blade model at the turning points, is it due to local or global buckling?

**Response to Comment 10:**

A linear buckling analysis of the blade is performed in order to investigate its buckling resistance and location of buckling eigenmodes. The results are depicted in Figure R 7. Negative eigenvalues correspond to the loads applied in the opposite direction, because no critical eigenvalue could be found in the load application direction. In other words, the blade exhibits sufficient buckling resistance for edgewise and combined edgewise and flap-wise loading. According to GL 2010 the load factor should be greater than 1.25, which is fulfilled for all the load cases studied. We observe that the eigenmodes are located in the sharp trailing edge structure for edgewise and combined edgewise and flap-wise loading cases. For the extreme flap-wise load case, eigenmode location is towards blade tip.

[Figure]

Figure R 7. Buckling modes of the blade under (a) 100% edgewise(min) loading case (b) 100% flap-wise(max) loading case (c) 100% combined edgewise(min) and flap-wise(max) loading. (Color bar shows total deformation).

According to Chen, Zhao und Xu (2017) a linear buckling analysis always predicts the upper borderline for the buckling factor, for a more realistic buckling analysis a nonlinear buckling analysis including geometric and material nonlinearity (degradation) is necessary.

After improving the mesh structure of the blade Fig. 24 (Figure R 8) is updated as seen below (See response to comment 8). In Figure R 8, turning points are marked for different load cases.

[Figure]

Figure R 8. Load displacement curves of the blade under edgewise, flap-wise, and edgewise plus flap-wise loading. (This figure updates Figure 24 in the original manuscript).

In Figure R 8 at the turning point which corresponds to 100% edgewise loading, the stiffness of the failed elements in suction side towards blade root are set to zero and total deformation due element failure in this area is observed as depicted Figure R 9 below.

[Figure]

Figure R 9. Failure evolution in the suction side of the blade at 100% edgewise loading (a) failed (killed) elements (b) total deformation due to failed elements.

For flapwise loading, first slight turning point due to the failure of elements in the internal flange is observed at 90% flap-wise load. At this load level, damage in the internal flange is shown in Figure R 5 (a). Second more obvious turning point is observed at 160% flap-wise loading. At this load level Figure R 5 (c) shows laminate failure in the internal flange and trailing edges. Due to element failure, deformation in the form of local buckling at the trailing edge is observed as depicted in Figure R 10 below.

For combined flap-wise and edgewise loading, similarly, first slight turning point due to the failure of elements in the internal flange is observed at 100% of combined loading. Damage in the internal flange is shown in Figure R 11 (a). Second more obvious turning point is observed under 170% combined flap-wise and edgewise loading. At this load level Figure R 11 (b) shows laminate failure in the internal flange, trailing edge and blade tip. Due to element failure, deformation in the form of local buckling at the trailing edge is observed as depicted in Figure R 11 below.

[Figure]

(a)

0    17.7770   35.5556   53.3333   71.1111   88.8889   106.667   124.444   142.222   160

(b)

Figure R 10. Failure evolution in the trailing edge of the blade at 160% flap-wise loading (a) failed (killed) elements (b) total deformation occurring in form of local buckling due to failed elements.

[Figure]

Figure R 11. Failure evolution in the pressure side, internal flange and suction side of the blade (from top to bottom in a row) at (a) 100% (b) 170 % combined edgewise and flap-wise loading (c) total deformation occurring in form of local buckling due to failed elements at 170 % combined edgewise and flap-wise loading.

Comment 11: In Fig. 19(c) and 19(d), why do the damaged regions heal? It is better to show the damage status rather than the Puck index. Like the one used in https://doi.org/10.3390/en7042274

**Response to Comment 11:**
After improving the mesh structure of the blade Fig. 16 (Figure R **12**) and Fig 19 (Figure R **13**) are updated as seen below (See response 8).

[Figure]

Figure R 12. Load displacement curves of the blade using the linear elastic model and Puck progressive model under edgewise loading. (This figure updates Figure 16 in the original manuscript).

In Fig. 19(c) and Fig 19(d) the dark blue regions inside red(damaged) regions do not mean that the damaged regions heal. Dark blue regions correspond to the 'killed' (failed) elements from previous load steps. Since the stiffness of the 'killed' elements are set to zero, they show no stress under loading. Hence their stress exposure (Puck's terminology for failure index) is calculated as zero and they appear as dark blue regions under red regions. For a more detailed explanation to this question please refer to the response of comment 9.

As recommended by the Referee the illustration of the damaged status is changed as depicted in Figure R 13 below:

[Figure]

Figure R 13. Failure progression on the blade pressure side, internal flange and suction side (from top to bottom in a row) at (a) 90%, (b) 100%, (c) 120% and (d) 150% of extreme edge-wise loading. (This figure updates Figure 19 in the original manuscript).

---

## Referee Comment (RC3) · Anonymous Referee #2 · 12 May 2020

In the paper, the authors investigate the static strength behavior of a 5 m wind turbine blade by means of finite element simulations. For this purpose, they utilize the well-known Puck failure criteria for the composite parts, both in a linear formulation and a non-linear degradation version. The linear version simply evaluates the strength criteria without stiffness degradation. The non-linear degradation version reduces the material stiffness in each ply whenever the Puck criteria are fulfilled and identifies laminate failure when three plies are failed. The authors analyze static extreme loads in flapwise, edgewise, and combined flap/edgewise directions and compare the results.

The topic of full-scale failure analysis of wind turbine rotor blades is generally interest-

ing and important for the wind energy research community. However, the manuscript does not represent a substantial, but rather a minor contribution to scientific progress within the scope of WES. The scientific approach and the applied methods are generally valid, but have weaknesses. At this point, the reviewer refers exemplarily to the incomplete iteration in the degradation model and the missing mesh convergence study (see below). The work is not reproducible, as there is no blade data available. The work can neither be repeated nor be verified by other scientists. The discussion of results is in parts not comprehensible, and does not include enough findings of other authors. The presentation quality is good in general. The text needs some revision due to typos and other minor language errors (that are too numerous to list).

In the following, the reviewer lists specific points of criticism that need to be addressed thoroughly in a potential revised version of the manuscript.

- The paper is an application paper rather than a research paper, as the utilized failure models are mainly already published elsewhere. The paper thus lacks novelty in the methodology development. The results are the novel content, but the basis for them, the blade design, is not publicly available due to confidentiality reasons. The added value of the paper for the research community is thus very limited. The fact that the results are neither reproducible nor verifiable is a strong weakness of the paper. A further weakness is the short length of the blade under investigation, which is not representative for modern wind turbines.

- Lines 16-17: The damage evolution is not necessarily linked to stiffness, as it is about stress and strength. It is easily possible to design stiff blades without damage, especially in the small size of the blade studied in this paper. The explanation via stiffness appears repeatedly throughout the text and should be discussed in more detail – or changed by more comprehensible argumentations.

- Equations (3) and (4): There are three equations, but only two numbers.

- Line 125 ff.: Why is the degradation model formulated in such a way, that the element

fails in case that 3 plies fail? Why 3 plies? That sounds unphysical to the reviewer. Wouldn't a relative number with respect to the overall number of plies be more meaningful? As after the modification of the stiffnesses the load is incremented, the authors do not perform a full iteration for the material degradation in each load step, which also seems unphysical or illogical from a numerical point of view. The authors should explain in more detail why the degradation model is formulated in this form.

- Figure 4: The reviewer suggests to change the line formats in order to highlight the own results: Black dots for experiments, red line with dots for the simulation.

- Line 161 ff.: The argumentation on the finite element mesh is weak from a scientific point of view. The mesh is actually quite coarse in some regions of the blade, which should be avoided. The reviewer recommends to perform a mesh convergence study, which should generally be done for finite element simulations, especially in science. The simulation time of 4 hours is nothing spectacular from the reviewer's point of view, so there is definitely room for further mesh refinement.

- Line 168: What is the reason for the limitation of the simulations to geometric linearity?

- Figure 5: The mesh looks weird in some regions. At the trailing edge, there are strange curves in the first element edges. One is well recognizable at the bottom of Fig. 5 (b). What is the reason for those?

- Figures 6-7: What is the coordinate system underlying the moment and force directions? Are the moments extreme for all positions, or just one position along the blade? Is it a mixture of different DLCs and time instances? Which DLC is the basis for the extreme loads?

- Figures 8-9: The load introduction is strange. Why didn't the authors use contact elements (MPCs), which is the standard way for distributed load introduction in 3D structures in ANSYS? The way the authors realize the load introduction may lead to spurious and erroneous local deformations. Please comment on that.

- Figures 10, 20, 24-26: Which load and deflection components are plotted? Which direction of load and deflection? Total load vs. total displacement? Which point exactly is traced in the deflection? It is just stated "a point close to the tip", which is imprecise.

- Figure 11: The quality of the text in the figure should be improved.

- Section 3 in general: The explanations of the results should be more precise. Why is the damage development as it is?

- Section 3.4: The reviewer does not understand why the combined loading is not the most severe one, as it should kind of add up the damage of flapwise and edgewise loading, especially in the linear part of Fig. 24. That holds for the entire section. The explanations have to be improved.

- Line 410: The authors state that the simulations have been carried out "before testing". Are tests planned? If so, it will be interesting to see if the simulations match well with the test.

- Conclusions, point 2.: Please add comprehensible explanations for the findings. Otherwise, there is not added value for the research community.

- Conclusions, point 3.: This finding is natural, as the fibers have the job to carry stresses and to provide stiffness. The stiffness contribution of the matrix is very limited – that's the nature of fiber composites.

- Conclusions, point 5.: How are the authors intending to increase the moment of inertia? By additional material or modifications in geometry?

- Conclusions, point 6.: The authors did not study the adhesives. How do they come to the conclusion that the main failure mechanism is expected to be linked to the adhesive joints?

- There are numerous language errors. No severe errors that make the paper unreadable, but the entire text needs revision.

- The reviewer highly recommends to make the blade data publicly available (including geometry and material layup), and also potential test results in the future. Otherwise, the value of the manuscript is, if present, very limited.
* * *

---

## Author Comment (AC3) · 29 May 2020

The authors would like thank the Referee for the valuable comments and detailed remarks regarding the numerical aspects in the manuscript. We believe that in light of the Referee comments our manuscript will become a more accurate and clear study with a greater impact. The comments are individually addressed below, with the Referee's comments written in red and our response in black.

In the paper, the authors investigate the static strength behavior of a 5 m wind turbine blade by means of finite element simulations. For this purpose, they utilize the well-known Puck failure criteria for the composite parts, both in a linear formulation and a non-linear degradation version. The linear version simply evaluates the strength criteria without stiffness degradation. The non-linear degradation version reduces the material stiffness in each ply whenever the Puck criteria are fulfilled and identifies laminate failure when three plies are failed. The authors analyze static extreme loads in flapwise, edgewise, and combined flap/edgewise directions and compare the results. The topic of full-scale failure analysis of wind turbine rotor blades is generally interesting and important for the wind energy research community. However, the manuscript does not represent a substantial, but rather a minor contribution to scientific progress within the scope of WES. The scientific approach and the applied methods are generally valid, but have weaknesses. At this point, the reviewer refers exemplarily to the incomplete iteration in the degradation model and the missing mesh convergence study (see below). The work is not reproducible, as there is no blade data available. The work can neither be repeated nor be verified by other scientists. The discussion of results is in parts not comprehensible, and does not include enough findings of other authors. The presentation quality is good in general. The text needs some revision due to typos and other minor language errors (that are too numerous to list).

Thank you for the general assessment in this comment regarding the manuscript.

First, we would like to emphasize the importance of the paper for the wind energy community as following:

According to the best knowledge of authors, this paper is the first one in literature concerning the FE Analysis of an existing small scale 5-m blade. It includes an in-depth detailed study of the potential composite failure patterns until collapse load using progressive Puck failure criteria which will be compared with tests that is planned to be conducted as a follow-up of this manuscript in the future.

Next, we would like to address here some of the general comments as follows which will be incorporated in the final revised version of the manuscript:

1. Geometry and material lay-up of the existing 5-meter METUWIND Blade will be shared with the community and will be included in the revised version.

2. Results and discussions will be presented in a more comprehensible and coherent style in the revised version. In some cases, the discussion of the results were already updated in AC1 and in AC2 as our response to RC1.

3. The literature will be extended to include the work and findings of other authors that was missed in our original manuscript and which will be added to the discussion of our results. Below is a list of the additional references that is added to the revised manuscript:

1. Paquette, J. A., & Veers, P. S. (2007). *Increased Strength in Wind Turbine Blades through Innovative Structural Design* (No. SAND2007-2632C). Sandia National Lab.(SNL-NM), Albuquerque, NM (United States).

2. Chen, X., Berring, P., Madsen, S. H., Branner, K., & Semenov, S. (2019). Understanding progressive failure mechanisms of a wind turbine blade trailing edge section through subcomponent tests and nonlinear FE analysis. *Composite Structures*, *214*, 422-438.

3. Chen, X., Qin, Z., Yang, K., Zhao, X., & Xu, J. (2015). Numerical analysis and experimental investigation of wind turbine blades with innovative features: Structural response and characteristics. *Science China Technological Sciences*, *58*(1), 1-8.

4.  Chen, X., Zhao, W., Zhao, X. L., & Xu, J. Z. (2014). Failure test and finite element simulation of a large wind turbine composite blade under static loading. *Energies*, *7*(4), 2274-2297.

5.  Fagan, E. M., Flanagan, M., Leen, S. B., Flanagan, T., Doyle, A., & Goggins, J. (2017). Physical experimental static testing and structural design optimisation for a composite wind turbine blade. *Composite Structures*, *164*, 90-103.

6.  Kim, S. H., Bang, H. J., Shin, H. K., & Jang, M. S. (2014). Composite structural analysis of flat-back shaped blade for multi-MW class wind turbine. *Applied Composite Materials*, *21*(3), 525-539.

7.  Montesano, J., Chu, H., & Singh, C. V. (2016). Development of a physics-based multi-scale progressive damage model for assessing the durability of wind turbine blades. *Composite Structures*, *141*, 50-62.

8.  Overgaard, L. C., Lund, E., & Thomsen, O. T. (2010). Structural collapse of a wind turbine blade. Part A: Static test and equivalent single layered models. *Composites Part A: Applied Science and Manufacturing*, *41*(2), 257-270.

9.  Sørensen, B. F., Jørgensen, E., Debel, C. P., Jensen, H. M., Jacobsen, T. K., & Halling, K. (2004). Improved design of large wind turbine blade of fibre composites based on studies of scale effects (Phase 1). Summary Report.

10. Yang, J., Peng, C., Xiao, J., Zeng, J., Xing, S., Jin, J., & Deng, H. (2013). Structural investigation of composite wind turbine blade considering structural collapse in full-scale static tests. *Composite Structures*, *97*, 15-29.

11. Zuo, Y., Montesano, J., & Singh, C. V. (2018). Assessing progressive failure in long wind turbine blades under quasi-static and cyclic loads. *Renewable Energy*, *119*, 754-766.

4. In the following you may find the italic text, which will be added to the manuscript concerning the findings of other authors. This text will be further extended in the revised version of the manuscript:

**Manuscript Line 344-346.** Element failure in the trailing edge begins at 110% loading. Similar to the pure flap-wise loading, number of failed elements increase in the trailing edge towards the spar and root as the load is further increased. *"In their study regarding the full-scale testing of a 34-m wind turbine blade, under combined loading Haselbach and Branner (2016) also observed laminate failure along the trailing edge."*
* * *
**Manuscript Line 256-259.** It is worth noting that, IFF(A) and IFF(B) do not lead to the element failure. When IFF(A) or IFF(B) occur, only the transverse, shear moduli and Poisson's ratio are reduced according the degradation rules.

**Manuscript Line 279-280.** We note FF and IFF(C) initiate in the same location as IFF(A) and/or IFF(B). *"Similar to our findings in Singh (2016)'s study IFF(A) and IFF(B) can be regarded as subcritical ply cracks which are precursor to more critical damage modes such as delamination. Since delamination was modeled within the scope of this study, IFF(C), which is a dangerous failure mode indicating delamination and FF are the critical failure modes which lead to element failure in this study."*
* * *
**Manuscript Section 3.1 (See Author Comment 2 (Response 8))**
At 180% load level trailing edge and the internal flange which is used to bond the pressure and suction sides of the blade are already damaged. As a consequence, towards the blade tip the pressure and suction sides of the blade are detached at this load level. Under these circumstances blade tip structure is weaker and can be damaged more easily. *"Likewise, debonding of suction and pressure sides from the adhesive joints was reported as the main failure mechanism causing a progressive collapse of the blade structure in Yang, Peng, Xiao, Zengi Xing, Jin and Deng (2013)."*
* * *
**Manuscript Section 3 discussion**
*"Laminate progression observed under flap-wise, edgewise and combined loading conditions fall into type 4 and type 5 wind blade damages as catagorized in Sorensen et al. (2014)."*
* * *
In the following, the reviewer lists specific points of criticism that need to be addressed thoroughly in a potential revised version of the manuscript.

- The paper is an application paper rather than a research paper, as the utilized failure models are mainly already published elsewhere. The paper thus lacks novelty in the methodology development. The results are the novel content, but the basis for them, the blade design, is not publicly available due to confidentiality reasons. The added value of the paper for the research community is thus very limited. The fact that the results are neither reproducible nor verifiable is a strong weakness of the paper. A further weakness is the short length of the blade under investigation, which is not representative for modern wind turbines.

**Response:**

In order to make the results reproducible for other researchers in the community, we will provide material lay-up and geometry in the revised manuscript and supplement. As RÜZGEM (METU Center for Wind Energy) is a partner research institute of EAWE, our policy is to collaborate and share information with researchers in wind energy.

The short length of the blade is attributed to the fact that the blade is constructed mainly for research purposes. Our primary goal is to investigate strength characteristics and failure mechanisms of the 5-meter blade using in-house developed software tools. Furthermore, building of affordable testing facilities are planned so that full-scale blade tests can be conducted and compared with simulations. We would like to point out that research wind turbine blades used by Paquette and Veers (2007) and Chen, Qin, Yang, Zhao and Xu (2014) were also short and only 9-meters and 10.3-meters respectively. Once the simulation results are compared with tests to be conducted in the future and the degree of agreement is assessed, we will be extending our study to the progressive failure analysis of large blades.

- Lines 16-17: The damage evolution is not necessarily linked to stiffness, as it is about stress and strength. It is easily possible to design stiff blades without damage, especially in the small size of the blade studied in this paper. The explanation via stiffness appears repeatedly throughout the text and should be discussed in more detail – or changed by more comprehensible argumentations.

-Section 3.4: The reviewer does not understand why the combined loading is not the most severe one, as it should kind of add up the damage of flapwise and edgewise loading, especially in the linear part of Fig. 24. That holds for the entire section. The explanations have to be improved. Figure 11: The quality of the text in the figure should be improved.

- Conclusions, point 2.: Please add comprehensible explanations for the findings. Otherwise, there is not added value for the research community.

**Response:**

We agree with the referee comments on the nature of the argument using stiffness and the stiffness argument is removed from the text. More comprehensible explanations in results and discussion section will be provided as explained below:
In combined loading due to the superposition of the loads 'stress state' inside the blade changes. Under pure edgewise loading the leading edge of the blade is subjected to compressive stresses, whereas flap-wise loading induces mainly tensile stresses in this area. In combined loading due to the effect of compressive stresses caused by edgewise loading, tensile stresses caused by flap-wise loading are reduced compared to the stress state in pure flap-wise loading case. This situation leads

to less element failure as depicted in Figure R 1. Figure R 1 shows the evolution of stress exposures IFF(C), FF in the blade at 100% pure flap-wise and combined loading stress states. As seen from the figure, for combined loading failed regions observed in the internal flange are less compared to pure flap-wise loading.

Please note that Figure R **1** is prepared after running the updated FE Model (See Author Comment 2 response 8).

[Figure]

Figure R 1. Failure evolution in the pressure side, internal flange and suction side of the blade (from top to bottom in a row) at 100 % (a) pure flap-wise (b) combined edgewise and flap-wise loading.

- Equations (3) and (4): There are three equations, but only two numbers.

**Response:**

Thank you for noticing the error.  The equations are renumbered as follows:

$$R_{\perp\perp}^A = \frac{S}{2p_{\perp\parallel}^{(-)}}\left[\sqrt{1 + 2p_{\perp\parallel}^{(-)}\frac{Y_c}{S}} - 1\right] \tag{3}$$

$$\tau_{21_c} = R_{\perp\parallel}\sqrt{1 + 2p_{\perp\perp}^{(-)}} \tag{4}$$

$$\text{and } p_{\perp\perp}^{(-)} = p_{\perp\parallel}^{(-)}\frac{R_{\perp\perp}^A}{S} \tag{5}$$

- Line 125 ff.: Why is the degradation model formulated in such a way, that the element fails in case that 3 plies fail? Why 3 plies? That sounds unphysical to the reviewer. Wouldn't a relative number with respect to the overall number of plies be more meaningful? As after the modification of the stiffnesses the load is incremented, the authors do not perform a full iteration for the material

degradation in each load step, which also seems unphysical or illogical from a numerical point of view. The authors should explain in more detail why the degradation model is formulated in this form.

**Response:**

In Passipoularidis, Philippidis and Brondsted (2016) (https://doi.org/10.1016/j.ijfatigue.2010.07.011). laminate failure takes place when IFF(C) is seen in all plies of laminate. Since IFF(C) is an explosive failure mode which indicates high risk of delamination, we wanted to implement a more conservative degradation scheme. As depicted in Figure R 2 our damage model delivers good agreement with experimental results. We agree with the Referee that a relative number with respect to overall number of plies would be more reasonable and we will modify our ANSYS APDL code accordingly. However, for our problem case, the laminate damage initiation and propagation begin at the leading edge and trailing edges where we have 9 plies. We have chosen the degradation scheme "if IFF(C) is observed in three plies or more element failure takes place" for these laminates. Regions with a greater number of plies are not damaged primarily. Therefore, we do not expect to significantly change our results.

Regarding carrying out iterations at each load steps, we have followed the flow chart of the program Subu in the book by Knops (2008) (ISBN 978-3-540-75765-8). According to the algorithm presented in the book, degradation is done per load step. In the revised version of the manuscript we will present our results with increased number of load steps, i.e. min 50 – 100 steps per load case as recommended in Knops (2008).

- Figure 4: The reviewer suggests to change the line formats in order to highlight the own results: Black dots for experiments, red line with dots for the simulation.

**Response:**

Fig.4 in the manuscript is changed according to the Referee's suggestion.

[Figure]

(a)  (b)

Figure R 2. Validation of the APDL Code for the progressive failure analysis of (a) $[0/90]_s$ GFRP/MY750 laminate under $\sigma_x$ uniaxial tension (b) $[0/\pm45/90]_s$ CFRP/AS4 3501-6 laminate under $\sigma_y$ uniaxial tension. (This figure updates Figure 4 in the original manuscript).

- Line 161 ff.: The argumentation on the finite element mesh is weak from a scientific point of view. The mesh is actually quite coarse in some regions of the blade, which should be avoided. The reviewer recommends to perform a mesh convergence study, which should generally be done for finite element simulations, especially in science. The simulation time of 4 hours is nothing spectacular from the reviewer's point of view, so there is definitely room for further mesh refinement.

**Response:**

New runs with a finer and corrected mesh was carried out in response to RC1 (See AC2) in which the new mesh is seen in Figure R 3. In addition, a mesh convergence study was carried out based on this final FE model as depicted in Figure R 4 which will be added to the revised manuscript. Mesh convergence is shown for total displacement at blade tip under extreme flap-wise loading and first eigenfrequency as seen in Figure R 4 (a) and (b) respectively. Current model (See Author Comment 2) contains a total number of 61104 elements with an element size 20x20 mm. This element size correlates to the number of elements used in the FE Modeling of small scale wind turbine blades in the literature. For a good compromise between accuracy and computation time in the revised manuscript version simulations with 101970 elements with an element size 15x15 mm will be presented.

[Figure]

Figure R 3. Mesh density used for the METUWIND Blade FE Model.

[Figure]

(a)                                              (b)

Figure R 4. Mesh convergence study using (a) total number of elements vs total displacement at blade tip (b) total number of elements vs 1. Eigenfrequency.

- Line 168: What is the reason for the limitation of the simulations to geometric linearity?

**Response**

We decided not to use the nonlinear geometry option in ANSYS, because under 100% flapwise, edgewise and combined loading conditions the total displacement of the blade is relatively small compared to the total blade length. Furthermore, as we are simulating full-scale testing until blade collapses, we wanted to avoid convergence problems at higher load levels and excessive computation time.

- Figure 5: The mesh looks weird in some regions. At the trailing edge, there are strange curves in the first element edges. One is well recognizable at the bottom of Fig. 5 (b). What is the reason for those?

**Response:**

We agree with the referee's observations. Mesh refinement was made and the mesh quality is improved. Please refer to Author Comment (AC) 2, response number 8 and figure R 3.

- Figures 6-7: What is the coordinate system underlying the moment and force directions? Are the moments extreme for all positions, or just one position along the blade? Is it a mixture of different DLCs and time instances? Which DLC is the basis for the extreme loads?

**Response:**

This question will be answered by referring to the Loads Report prepared by SMART BLADES GmbH (Weinzierl and Pechlivanoglou, 2013). The forces and moments are given in the blade pitch coordinate system according to GL Guidelines 2010.

[Figure]

Figure R 5. Blade pitch coordinate system.

Moments are extreme for all positions along the blade. Mixture of extreme load cases and time instances used in the study are listed below in Table R 1:

Table R 1. List of extreme load cases used.

| $LC_{index}$ | File name of load case |
| --- | --- |
| 1 | DLC5.1_EWM50_59.50ms_180.0degYAW_0.00degV_final_Trf.out |
| 2 | DLC5.1_EWM50_59.50ms_0.0degYAW_-8.00degV_final_Trf.out |
| 3 | DLC2.3_EOG1_19.00ms_8.00degH_-8.00degV_Trf.out |
| 4 | DLC5.1_EWM50_59.50ms_-135.0degYAW_0.00degV_final_Trf.out |
| 5 | DLC2.1_NWP_21.25ms_8.0degV_-45.0degYAW_120.0rpm_final_Trf.out |
| 6 | DLC1.2_ECD_11.00ms_0.00degH_-8.00degV_pos_final_Trf.out |
| 7 | DLC1.2_ECD_14.00ms_0.00degH_8.00degV_neg_final_Trf.out |
| 8 | DLC2.3_EOG1_3.00ms_-8.00degH_-8.00degV_Trf.out |
| 9 | DLC2.3_EOG1_3.00ms_-8.00degH_8.00degV_Trf.out |
| 10 | DLC2.3_EOG1_5.00ms_-8.00degH_-8.00degV_Trf.out |
| 11 | DLC1.4_EDC50_13.00ms_8.00degH_0.00degV_pos_final_Trf.out |
| 12 | DLC1.2_ECD_7.00ms_8.00degH_-8.00degV_neg_final_Trf.out |
| 13 | DLC2.3_EOG1_3.00ms_8.00degH_8.00degV_Trf.out |
| 14 | DLC5.1_EWM50_59.50ms_180.0degYAW_-8.00degV_final_Trf.out |
| 15 | DLC1.3_EOG50_3.00ms_-8.00degH_0.00degV_final_Trf.out |
| 16 | DLC1.4_EDC50_21.00ms_0.00degH_-8.00degV_neg_final_Trf.out |
| 17 | DLC2.1_NWP_21.25ms_0.0degV_180.0degYAW_120.0rpm_final_Trf.out |
| 18 | DLC2.3_EOG1_11.00ms_-8.00degH_8.00degV_Trf.out |
| 19 | DLC1.2_ECD_13.00ms_0.00degH_0.00degV_pos_final_Trf.out |
| 20 | DLC1.4_EDC50_9.00ms_0.00degH_0.00degV_neg_final_Trf.out |
| 21 | DLC1.4_EDC50_21.00ms_-8.00degH_-8.00degV_neg_final_Trf.out |
| 22 | DLC2.3_EOG1_19.00ms_8.00degH_8.00degV_Trf.out |
| 23 | DLC1.2_ECD_14.00ms_0.00degH_0.00degV_neg_final_Trf.out |
| 24 | DLC2.1_NWP_21.25ms_0.0degV_-45.0degYAW_120.0rpm_final_Trf.out |
| 25 | DLC2.1_NWP_21.25ms_8.0degV_-135.0degYAW_120.0rpm_final_Trf.out |
| 26 | DLC2.3_EOG1_7.00ms_-8.00degH_0.00degV_Trf.out |
| 27 | DLC1.2_ECD_11.00ms_-8.00degH_-8.00degV_neg_final_Trf.out |
| 28 | DLC1.2_ECD_9.00ms_8.00degH_8.00degV_neg_final_Trf.out |
| 29 | DLC2.3_EOG1_19.00ms_8.00degH_0.00degV_Trf.out |
| 30 | DLC2.3_EOG1_19.00ms_0.00degH_0.00degV_Trf.out |
| 31 | DLC1.2_ECD_14.00ms_-8.00degH_8.00degV_neg_final_Trf.out |

**-** Figures 8-9: The load introduction is strange. Why didn't the authors use contact elements (MPCs), which is the standard way for distributed load introduction in 3D structures in ANSYS? The way the authors realize the load introduction may lead to spurious and erroneous local deformations. Please comment on that.

**Response:**

Our load introduction approach, where the load is distributed along spar width at suction and pressure side and the approach proposed by the Referee are both available in literature. When the short dimensions of the blade are considered, we do not believe that changing our load introduction methodology to MPC will significantly affect our results.

- Figures 10, 20, 24-26: Which load and deflection components are plotted? Which direction of load and deflection? Total load vs. total displacement? Which point exactly is traced in the deflection? It is just stated "a point close to the tip", which is imprecise.

**Response:**

Please note the following explanations for the figures:

Figure 10:
Total Flap-wise loading component, total deflection in flap-wise direction

Figure 20:
Total resultant combined (Flap-wise+Edgewise) loading component, total deflection

Figure 24:
Total Flap-wise loading component, total deflection in flap-wise direction
Total Edge-wise loading component, total deflection in edgewise direction
Total resultant combined (Flap-wise+Edgewise) loading component, total deflection

Figure 25:
 Total Flap-wise loading component, total deflection in flap-wise direction
 Total resultant combined (Flap-wise+Edgewise) loading component, deflection component in flap-wise direction

Figure 26:
Total Edge-wise loading component, total deflection in edgewise direction
Total resultant combined (Flap-wise+Edgewise) loading component, deflection component in edgewise direction

The exact location and coordinates of the deflection measurement point close to blade tip is highlighted in green in Figure R 6. Point coordinates are x= 38.493 y=4750. z-31.51 with respect to the coordinate system located at blade root.

[Figure]

Figure R 6. Location of the deflection measurement point.

**-** Figure 11: The quality of the text in the figure should be improved.

**Response:**

As suggested by the Referee, the quality of the Figure text is improved and will be used in the revised version.

- Section 3 in general: The explanations of the results should be more precise. Why is the damage development as it is?

**Response**

Detailed and precise explanation was done in Authors Comment (AC) 2 response number 8 for extreme flap-wise loading. Similar explanations will be done for edgewise and combined loading conditions in the revised manuscript.

**-** Line 410: The authors state that the simulations have been carried out "before testing". Are tests planned? If so, it will be interesting to see if the simulations match well with the test.

**Response:**

In the first part of our project, progressive failure analysis of the existing 5-meter METUWIND blade is carried out as presented in this manuscript. In the second part of our research project, tests for the existing blade seen in Figure R 7 will be conducted following the completion of the test fixture. Afterwards, simulation results from this manuscript will be compared with tests.

[Figure]

Figure R 7. Picture of the 5-meter METUWIND Blade.

- Conclusions, point 3.: This finding is natural, as the fibers have the job to carry stresses and to provide stiffness. The stiffness contribution of the matrix is very limited
– that's the nature of fiber composites.

**Response:**

We agree with the Referee, and the Conclusion, point 3 will be reworded in the revised version of the manuscript.

- Conclusions, point 5.: How are the authors intending to increase the moment of inertia? By additional material or modifications in geometry?

- Conclusions, point 6.: The authors did not study the adhesives. How do they come to the conclusion that the main failure mechanism is expected to be linked to the adhesive joints?

**Response:**

Conclusions, point 5 and Conclusions, point 6 will be deleted in the revised version of the manuscript.

- There are numerous language errors. No severe errors that make the paper unreadable, but the entire text needs revision.

**Response:**

We have corrected numerous language errors in the revised version of the original manuscript (See Author Comment AC1 on WES Discussion page):

https://www.wind-energ-sci-discuss.net/wes-2020-44/wes-2020-44-AC1-supplement.pdf

We will correct language and typo errors in the revised version.

**-** The reviewer highly recommends to make the blade data publicly available (including geometry and material layup), and also potential test results in the future. Otherwise, the value of the manuscript is, if present, very limited.

**Response:**

As mentioned in Response to Comment 1 CAD data of the blade components, i.e. suction side, pressure side, internal flange and 'hat shaped' spar will be provided. Moreover, lamination plan of the blade suction side, pressure side, internal flange and 'hat shaped' spar will be shared.

---

## Author Response (AR1)

The authors would like thank the Referees for the detailed and insightful comments regarding the manuscript. We have revised the manuscript in light of the Referee comments as we individually pointed out in our response.  Please note the following main points in the revised manuscript:

- We have incorporated the individual responses to the Referee comments to the revised manuscript by carrying out the necessary modifications and adding new figures and text where necessary. These changes are marked in yellow in the revised manuscript in attachment.

- Geometry and material lay-up of the existing 5-meter RUZGEM Blade is shared with the community in the supplement.

[revised manuscript text omitted]

---

## Author Response (AR2)

We would like thank the Referees and the Associate Editor for their evaluation and feedback regarding our revised manuscript. We have addressed the two points according to the associate editor's suggestions for the improvement of the manuscript. These comments are individually addressed below with the Referee's comments written in red and our response in black.

Comment 1: Please explain in detail why combined loading leads to less damage than pure flap-wise loading.

Response to Comment 1:

We have changed the text and expanded it to more clearly explain why combined load leads to less damage. We have reconstructed the table 3 and Fig. 29 to support the related arguments in the discussion section. Our argument is highlighted in yellow (lines 466-483 and lines 546-548) in the marked-up manuscript version.

Comment 2: Please add one paragraph to describe the limitations, if any, of the present study and some improvement may be included in the future study.

Response to Comment 2:

The summary of the limitations in the study and improvement for future work is added in the method and conclusion sections. In the marked-up manuscript, the highlighted lines 187-191 and 551-553 in yellow have been added for the limitations of the study. Likewise, the highlighted lines 185-186 and 553-557 in yellow have been added for future work.

Additional modifications done in the revised manuscript are as follows:

- Typo errors are corrected as shown in the marked version of the revised manuscript
- Missing reference for Fig. 4 (WWFE, I, Hinton et al., 2002) is added
- Missing Linear elastic load-displacement curve is added to Fig. 15

Our above-mentioned additional modifications are highlighted in yellow in the marked-up manuscript version below.

[revised manuscript text omitted]